



# Inter-comparison of CO measurements from TROPOMI, ACE-FTS, and a high-Arctic ground-based FTS

Tyler Wizenberg[1], Kimberly Strong[1], Kaley A. Walker[1], Erik Lutsch[1], Tobias Borsdorff[2], and Jochen Landgraf[2]

[1]Department of Physics, University of Toronto, Toronto, Ontario, Canada
[2]SRON Netherlands Institute for Space Research, Utrecht, The Netherlands

**Correspondence:** Tyler Wizenberg (wizenberg@atmosp.physics.utoronto.ca)

**Abstract.** The TROPOspheric Monitoring Instrument (TROPOMI) provides a daily, spatially-resolved (initially $7 \times 7 \text{ km}^2$, upgraded to $7 \times 5.6 \text{ km}^2$ in August 2019) global data set of CO columns, however, due to the relative sparseness of reliable ground-based data sources, it can be challenging to characterize the validity and accuracy of satellite data products in remote regions such as the high Arctic. In these regions, satellite inter-comparisons can supplement model- and ground-based valida-

tion efforts and serve to verify previously observed differences. In this paper, we compare the CO products from TROPOMI, the Atmospheric Chemistry Experiment (ACE) Fourier Transform Spectrometer (FTS), and a high-Arctic ground-based FTS located at the Polar Environment Atmospheric Research Laboratory (PEARL) in Eureka, Nunavut (80.05° N, 86.42° W).

A global comparison of TROPOMI reference profiles scaled by the retrieved total column with ACE-FTS CO partial columns for the period from 10 November 2017 to 31 May 2020 displays excellent agreement between the two data sets ($R = 0.93$),

and a small relative bias of $-0.68 \pm 0.25$ % (bias $\pm$ standard error). Additional comparisons were performed within five latitude bands; the north Polar region (60° N to 90° N), northern Mid-latitudes (20° N to 60° N), the Equatorial region (20° S to 20° N), southern Mid-latitudes (60° S to 20° S), and the south Polar region (90° S to 60° S). Latitudinal comparisons of the TROPOMI and ACE-FTS CO data sets show strong correlations ranging from $R = 0.93$ (southern Mid-latitudes) to $R = 0.85$ (Equatorial region) between the CO products, but display a dependence of the mean differences on latitude. Positive

mean biases of $7.92 \pm 0.58$ % and $7.98 \pm 0.51$ % were found in the northern and southern Polar regions, respectively, while a negative bias of $-9.16 \pm 0.55$ % was observed in the Equatorial region. To investigate whether these differences are introduced by cloud contamination which is reflected in the TROPOMI averaging kernel shape, the latitudinal comparisons were repeated for cloud-covered pixels and clear-sky pixels only, and for the unsmoothed and smoothed cases. Clear-sky pixels were found to be biased higher with poorer correlations on average than clear+cloudy scenes and cloud-covered scenes only. Furthermore,

the latitudinal dependence on the biases was observed in both the smoothed and unsmoothed cases.

To provide additional context to the global comparisons of TROPOMI with ACE-FTS in the Arctic, both satellite data sets were compared against measurements from the ground-based PEARL-FTS. Comparisons of TROPOMI with smoothed PEARL-FTS total columns in the period of 3 March 2018 to 27 March 2020 display a strong correlation ($R = 0.88$), however a positive mean bias of $14.3 \pm 0.16$ % was also found. A partial column comparison of ACE-FTS with the PEARL-FTS

in the period from 25 February 2007 to 18 March 2020 shows good agreement ($R = 0.82$), and a mean positive bias of



9.83 ± 0.22 % in the ACE-FTS product relative to the ground-based FTS. The magnitude and sign of the mean relative differences are consistent across all inter-comparisons in this work, as well as with recent ground-based validation efforts, suggesting that current TROPOMI CO product exhibits a positive bias in the high-Arctic region. However, the observed bias is within the TROPOMI mission accuracy requirement of ± 15 %, providing further confirmation that the data quality in these

remote high-latitude regions meets this specification.

# 1 Introduction

Carbon monoxide (CO) is an atmospheric pollutant that is present in relatively low concentrations globally, but affects air quality, the climate, and human health. In the troposphere, CO is primarily produced through incomplete combustion from

both anthropogenic and natural sources, including industrial activities, heating, and biomass burning (van der Werf et al., 2010; Granier et al., 2011). CO has an indirect influence on the global budgets of greenhouse gases such as $CH_4$, $CO_2$, and $O_3$, and it serves as an important sink of the hydroxyl radical (OH), having implications for the chemistry and oxidative power of the atmosphere (Logan et al., 1981; Spivakovsky et al., 2000; Lelieveld et al., 2016). In the remote high-Arctic region, local emissions of CO are negligible, and transported anthropogenic and biomass burning emissions are the primary sources of CO

and other reactive species (Yurganov, 1997; Law et al., 2014; Lutsch et al., 2020). Due to its long lifetime in the atmosphere of approximately two months, CO serves as an important long-range tracer species for observations of wildfire pollution plumes, particularly in these remote regions (Duflot et al., 2010; Lutsch et al., 2016, 2019). As a result, having accurate and reliable measurements of CO columns over the high Arctic can enable quantification of the role and impacts of biomass burning pollution on the Arctic climate and environment.

Satellite-borne remote sensing instruments are valuable tools for global observations of CO concentrations and distribution. However, the validation of such instruments over remote areas such as the Arctic, where ground-based measurements are sparse, can be challenging. The TROPOspheric Monitoring Instrument (TROPOMI) provides the highest spatially resolved measurements of CO from space currently available, and is extending the global CO record established by previous satellite instruments including Measurements of Pollution In The Troposphere (MOPITT; Drummond and Mand, 1996), the Atmo-

spheric Infrared Sounder (AIRS; Chahine et al., 2006) , the Infrared Atmospheric Sounding Interferometer (IASI; Clerbaux et al., 2009), and the Cross-track Infrared Sounder (CrIS; Han et al., 2013). Since the launch of TROPOMI in October 2017, the operational CO product has been validated against the European Centres for Medium Range Weather Forecasting (ECMWF) Integrated Forecasting System (IFS) in Borsdorff et al. (2018a), against ten ground-based stations of the Total Carbon Column Observing Network (TCCON; Wunch et al., 2011) in Borsdorff et al. (2018b), and most recently against twenty eight TC-

CON stations and twenty two ground-based stations of the Network for the Detection of Atmospheric Composition Change (NDACC; De Mazière et al., 2018) in Sha et al. (2021). The recent ground-based validation work by Sha et al. (2021) was



the first to incorporate measurements from high-Arctic sites including Eureka, Nunavut (80.05° N, 86.42° W), Ny-Ålesund, Svalbard (78.90° N, 11.90° E), and Thule, Greenland (76.52° N, 68.77° W). Higher-than-average biases were observed in the operational TROPOMI CO product of: $12.96 \pm 4.56$ %, $11.72 \pm 3.82$ %, and $9.44 \pm 4.79$ % against NDACC measurements,

and $6.4 \pm 4.18$ %, $7.54 \pm 4.4$ %, and $5.75 \pm 4.93$ % against TCCON measurements for Eureka, Ny-Ålesund, and Thule respectively. Updates to the retrieval spectroscopy and de-striping algorithm methodology proposed in Borsdorff et al. (2019) appear to ameliorate the positive bias observed at high latitude sites, but this improved CO product is not yet available. Satellite inter-comparisons are complementary to model- and ground-based validation efforts, and can serve to verify previously observed differences, particularly where ground-based measurements are limited. Furthermore, inter-comparisons such as these

can help to position newer instruments in the context of the measurement record of preceding instruments. Currently, the only satellite-borne instrument that the TROPOMI CO product has been directly compared against is MOPITT in Martínez-Alonso et al. (2020), and thus inter-comparisons with additional satellite data sources are important.

The Atmospheric Chemistry Experiment - Fourier Transform Spectrometer (ACE-FTS; Bernath et al., 2005) is currently the only solar occultation limb-measuring instrument in orbit that is capable of retrieving high-vertical-resolution atmospheric

profiles of CO. ACE-FTS is well-validated over the high-Arctic region, and it has been involved in the yearly Canadian Arctic ACE-OSIRIS Validation Campaigns since Spring 2004. Earlier versions of the ACE-FTS CO product have been validated against both satellite and ground-based (including high-Arctic) measurements from NDACC namely by Clerbaux et al. (2008), Griffin et al. (2017), and Sheese et al. (2017). ACE-FTS profiles have previously been employed for the validation of $CH_4$ measurements from the nadir-sounding TANSO-FTS instrument aboard GOSAT over the Arctic in Holl et al. (2016) and

Olsen et al. (2017). Due to their differing orbits, TROPOMI and ACE-FTS benefit from the highest degree of overlap in their measurements at the north and south Polar regions, providing a unique opportunity for an inter-comparison of these two data products in these remote high-latitude regions. Through the inclusion of correlative high-spectral-resolution ground-based NDACC measurements made at the Polar Environment Atmospheric Research Laboratory (PEARL) located in Eureka, Nunavut (the northernmost NDACC station), we gain additional context and a baseline standard to which the two satellite

instruments can be compared. Here, we perform a global comparison of collocated ACE-FTS and TROPOMI measurements, as well as a localized comparison of both satellite instruments with high-Arctic ground-based Fourier transform spectrometer (FTS) measurements made at PEARL, with the goal of highlighting any latitudinal trends or features in the TROPOMI product, and to further characterize the accuracy of high latitude TROPOMI CO measurements.

This paper is structured as follows: the various datasets used in this study are described in Sect. 2, and the methodology used

for comparing each instrument are described in Sect. 3. The results of the comparisons are presented and discussed in Sect. 4, and the conclusions are provided in Sect. 5.





## 2  Datasets

### 2.1  TROPospheric Monitoring Instrument (TROPOMI)

TROPOMI is the exclusive payload aboard the European Space Agency's Sentinel 5-Precursor (S5-P) satellite, which was
launched on 13 October 2017 into a high-inclination (98.7°), sun-synchronous orbit at an altitude of 824 km, with a 13:30 local
standard time Equator crossing time (Veefkind et al., 2012). The TROPOMI instrument is a nadir-viewing push-broom grating
spectrometer array, comprised of four individual spectrometers (UV-Vis-NIR-SWIR), with a swath width of 2600 km, and a
$7.2 \times 7.2 \, \mathrm{km}^2$ footprint at nadir for CO (Veefkind et al., 2012). The footprint at nadir was further reduced to $7 \times 5.6 \, \mathrm{km}^2$ from
6 August 2019 on-wards through improvements to the electronic read-out rate of the spectrometer analog-to-digital converter.
For CO, total column densities are obtained from Earth radiance spectra in the shortwave IR spectral window around 2.3 μm,
where the first overtone absorption band of CO is located. Retrievals over land are performed for both clear-sky and cloudy
conditions, however retrievals over oceans and other large bodies of water are only possible during cloudy conditions due to
the low reflectivity of open water (Landgraf et al., 2016). The current TROPOMI CO processor uses spectroscopic parameters
from the HITRAN 2008 line-list database (Rothman et al., 2009) with updates to the water vapor spectroscopy (Scheepmaker
et al., 2013).

Vertically integrated CO column densities are retrieved from TROPOMI's shortwave infrared measurements using the Short-
wave Infrared Carbon Monoxide Retrieval (SICOR) algorithm, which was developed specifically for the S5-P and S5 missions
(Vidot et al., 2012). The SICOR retrieval algorithm employs a profile-scaling approach whereby the retrieval state vector
contains a single scaling factor that represents the ratio of the retrieved CO total column to the *a priori* CO total column abun-
dance. The *a priori* reference profiles are generated from the TM5 3D global chemical transport model (Krol et al., 2005), and
they vary based on location, month, and year. Thus, the final retrieved CO total column density corresponds to the vertically-
integrated scaled reference profile (Landgraf et al., 2016). The shape of the column averaging kernels of the CO retrievals
varies based on the cloud fraction of a given measurement, reflecting the sensitivity loss of the retrieval due to cloud contam-
ination. In general, for clear-sky retrievals over land, the averaging kernel of the SICOR retrieval is near unity for the entire
vertical extent of the profile, meaning that it is relatively insensitive to the vertical distribution of CO. However, for retrievals
performed in the presence of cloud fractions greater than 0, the column averaging kernel values will decrease towards zero in
the region below the clouds, while simultaneously increasing to values larger than one above the cloud, leading to an increased
sensitivity to the CO partial column above the height of the clouds (Landgraf et al., 2016). This approach compensates for the
effects of cloud shielding on the retrieved CO column, however for retrievals made in these conditions, the shape of the *a priori*
profiles may introduce some additional error into the retrieved total columns (Borsdorff et al., 2014). The mission accuracy and
precision requirements for TROPOMI's CO product are 15 % and 10 %, respectively (Landgraf et al., 2016). Further details
on the TROPOMI CO retrieval algorithm can be found in Landgraf et al. (2016).

In this work, we analyze TROPOMI CO measurements for the period from 10 November 2017 to 31 May 2020. We use either
the reprocessed (RPRO) or offline (OFFL) data files from the most recent processor versions (010001, 010002, 010202, 010300,
010301, and 010302) depending on availability for a given day of observations. Individual pixels are filtered using the quality

flag variable ("qa_value"), which is a discrete value that provides a quality percentage (Landgraf et al., 2018). Pixels with a qa_value < 0.5 are removed prior to analysis as suggested in the algorithm theoretical baseline document (ATBD) (Landgraf et al., 2018). Furthermore, the quality values were also used to differentiate clear-sky scenes (qa_value = 1.0, representing an optical thickness < 0.5 and cloud height < 500 m) from cloudy scenes ($0.5 \leq$ qa_value $\leq 0.7$, representing an optical thickness

$\geq 0.5$ and cloud height < 5000 m) for later analysis, as described in the CO product read-me file (Landgraf et al., 2020).

## 2.2 ACE-FTS

ACE-FTS was launched on board the Canadian Space Agency's SCISAT satellite into a low-Earth circular orbit at an altitude of 650 km and an inclination of 74° on 12 August 2003. This orbit provides ACE with latitudinal coverage between 85° and −85° (Bernath et al., 2005). The FTS is the primary instrument aboard SCISAT, but it is also accompanied by Measurement

of Aerosol Extinction in the Stratosphere and Troposphere Retrieved by Occultation (MAESTRO), a dual spectrophotometer primarily aimed at improving our understanding of polar ozone chemistry (McElroy et al., 2007). In this work, we focus solely on measurements from ACE-FTS.

ACE-FTS is an infrared Michelson interferometer which was designed and constructed by ABB Inc. in Quebec City, Canada. It has a high spectral resolution of $0.02 \, \mathrm{cm}^{-1}$, and it covers the wavenumber range between 750-4440 $\mathrm{cm}^{-1}$. ACE-FTS makes

up to 30 measurements per day by solar occultation at sunrise and sunset, and provides vertical profile information (typically between 5-110 km) of temperature, pressure and volume mixing ratios (VMR) for 68 molecules and isotopologues in the most recent data version (v4.1) (Boone et al., 2020). ACE-FTS has a variable vertical sampling of 1.5-6 km, and a mean vertical resolution of ∼3-4 km, which varies based on the orbit, beta angle, and instrument field-of-view (Boone et al., 2005).

CO VMR profiles from the latest version of the ACE-FTS data (v4.1) are used in this study (Boone et al., 2020). The VMR

profiles are retrieved from the measured infrared spectra using a global-fit algorithm which employs a Levenburg-Marquardt non-linear least-squares fitting method as described in Boone et al. (2005). ACE-FTS L2 data are provided in two varieties: one that is on the original retrieval altitude grid, and another that has been interpolated onto a fixed 1-km grid. Here, we use only the version with the 1-km grid. Individual ACE-FTS occultations are filtered using the quality flags, following the suggestions provided in Sheese et al. (2015). Furthermore, to maximize the vertical information coming from ACE-FTS, we

discard retrieved profiles with an excessive number of fill values (i.e., missing data), and those for which the lowest measured altitude is above 10.5 km.

## 2.3 PEARL-FTS

The ground-based instrument used in this study is a Bruker IFS 125HR Fourier transform spectrometer located at the Polar Environment Atmospheric Research Laboratory Ridge Laboratory (80.05° N, 86.42° W; 610 m ASL) in Eureka, Nunavut,

Canada (Batchelor et al., 2009). The PEARL Ridge Laboratory is operated by the Canadian Network for the Detection of Atmospheric Change (CANDAC), and is situated approximately 15 km away from the Environment and Climate Change Canada (ECCC) Eureka Weather Station (79.98° N, 85.93° W; 0 m ASL) (Fogal et al., 2013). The PEARL Ridge Lab is a remote site, and is minimally influenced by local pollution sources. The PEARL-FTS was installed in July 2006, and has





been involved in the annual Canadian Arctic ACE-OSIRIS Validation Campaigns held during polar sunrise since Spring 2007,
and has been previously compared with ACE-FTS and other satellite-borne instruments, for example: Clerbaux et al. (2008),
Batchelor et al. (2010), Holl et al. (2016), Buchholz et al. (2017), Griffin et al. (2017), Olsen et al. (2017), Bognar et al. (2019),
Weaver et al. (2019), and Vigouroux et al. (2020).

The PEARL-FTS is a high-spectral-resolution ($0.0035$ cm$^{-1}$) Michelson interferometer produced by Bruker Optics. Using
a custom-built solar-tracker system and the sun as a source, it makes atmospheric solar-absorption measurements in the mid-
infrared region between 600-4300 cm$^{-1}$ during the sunlit portion of the year (Batchelor et al., 2009). The interferograms are
collected using one of two liquid-nitrogen cooled detectors; a photoconductive mercury-cadmium-telluride (HgCdTe) detector
or a photovoltaic indium-antimonide (InSb) detector. Additionally, seven internal narrow-bandpass filters are used, which
limit the wavenumber range of the measured spectra, thus increasing the signal-to-noise ratio (SNR) (Batchelor et al., 2009).
The instrument is part of NDACC (http://www.ndaccdemo.org/about; De Mazière et al., 2018), and measurements of CO,
CH$_4$, and O$_3$ are regularly provided to the Copernicus Atmospheric Monitoring Service (CAMS) rapid delivery initiative. In
addition, the instrument is capable of near-infrared measurements using a third indium-gallium-arsenide (InGaAs) detector,
and observations in the near-IR are contributed to the TCCON (Wunch et al., 2011). In this work, however, only the NDACC
mid-infrared measurements of CO are used.

From the measured solar absorption spectra, vertical profiles and total and partial column trace-gas abundances are retrieved
using the SFIT4 v0.9.4.4 retrieval software (https://wiki.ucar.edu/display/sfit4/) which is based upon the Optimal Estimation
Method (OEM) of Rodgers (2000). The SFIT4 retrieval algorithm iteratively fits a calculated spectrum to the observed spec-
tra by adjusting the VMR profile of the target gas on a 47-layer vertical grid (extending from 0.61 km (the altitude of the
Ridge Lab) to 120 km) until a convergence criterion is met. For the retrieval of CO, the microwindows and interfering species
recommended by NDACC were used (Table 1). The OEM retrieval procedure requires prior knowledge of the atmosphere
as input, including daily atmospheric profiles of pressure and temperature from the US National Centers for Environmental
Prediction (NCEP, ftp://ftp.cpc.ncep.noaa.gov/ndacc/ncep/) interpolated to the location of PEARL, and *a priori* trace-gas pro-
files that are sourced from a 40-year average (1980-2020) of the Whole Atmosphere Community Climate Model (WACCM,
https://www2.acom.ucar.edu/gcm/waccm) v4 for Eureka (Marsh et al., 2013). Above the 10 Pa pressure level (~45 km) NCEP
P-T profiles are unavailable, so in this region the mean pressure and temperature profiles from the aforementioned WACCM
model run are used. Additionally, spectroscopic parameters used in the retrieval process for CO are from ATM16 (Toon, 2015),
while all other species are from HITRAN 2008 (Rothman et al., 2009).



**Table 1.** NDACC CO microwindows and interfering species used in SFIT4 V0.9.4.4 retrievals for the PEARL-FTS.

| Microwindow # | Wavenumber range ($cm^{-1}$) | Interfering species |
|---|---|---|
| 1 | 2057.70-2058.00 | $CO_2$, $O_3$, OCS |
| 2 | 2069.56-2069.76 | $CO_2$, $O_3$, OCS |
| 3 | 2157.50-2159.15 | $CO_2$, $O_3$, OCS, $N_2O$, $H_2O$ |

## 3   Methods

### 3.1   Collocations and averaging

In this study, we consider a pair of instruments to be collocated when they are observing the same approximate airmass, at the same approximate time. For the comparisons presented here, broad collocation criteria of 24 hours in time, and 500 km in space were used to maximize the quantity of data available. A range of stricter collocation criteria were tested, but no significant trend between the inter-instrument differences and the spatial and temporal collocation criteria was found. Similarly broad collocation criterion were used in previous ACE-FTS CO validation studies by Clerbaux et al. (2008) and Griffin et al. (2017).

In the determination of collocated measurements, we consider each ACE-FTS profile as a point measurement, using the geographical location of the 30-km tangent-point for the calculation of the inter-instrument distances. For comparisons involving the PEARL-FTS, we use the location of the PEARL Ridge Laboratory. It should be noted that for both ACE-FTS and the PEARL-FTS, these measurements do not occur at a single point, but rather along a broad horizontal slant path through the atmosphere. Drawing upon the example provided in Holl et al. (2016), for a limb-sounding measurement with a 10-km tangent height, the horizontal extent of the slant path is approximately 715 km in the altitude range of 10-50 km.

For the comparison of ACE-FTS and TROPOMI, collocations between the two instruments occur globally, spanning the latitudinal range of 82° N to 81° S. For comparisons involving the PEARL-FTS, collocations are limited to the geographical area within a great-circle radius of 500 km surrounding the PEARL Ridge Laboratory. A summary of the collocation statistics for each instrument pair is provided in Table 2.

Due to the broad swath width of TROPOMI, a single ACE-FTS or PEARL-FTS measurement can collocate with thousands of TROPOMI pixels at once. As a result, for the comparisons of TROPOMI with ACE-FTS and the PEARL-FTS, we compute the arithmetic average of all collocated TROPOMI pixels for each ACE-FTS or PEARL-FTS observation. A similar approach was applied in the comparisons of $CH_4$ measurements from ACE-FTS and the nadir-sounder TANSO-FTS onboard GOSAT in Holl et al. (2016), as well as in De Mazière et al. (2008). To ensure the statistical robustness of the averaging, collocations with a small number of pixels ($< 50$) are removed prior to analysis. These cases displayed significantly larger variances than those with a large number of pixels. For comparisons of ACE-FTS to the PEARL-FTS, no averaging was applied, and a single ACE-FTS profile was allowed to collocate with multiple PEARL-FTS measurements and vice-versa.





**Table 2.** Summary of the collocation statistics for each pair of instruments. Collocations between TROPOMI and ACE-FTS occur globally, while collocations involving the PEARL-FTS are limited to the region within a 500 km radius from the Ridge Laboratory. The uncertainties provided for the mean distances and times are the standard deviations.

| Primary instrument | PEARL-FTS | PEARL-FTS | ACE-FTS |
|---|---|---|---|
| Secondary instrument | ACE-FTS | TROPOMI | TROPOMI |
| First collocation | 25 February 2007 | 3 March 2018 | 10 November 2017 |
| Last collocation | 18 March 2020 | 27 March 2020 | 31 May 2020 |
| No. collocations | 2096 | 1875 | 6136 |
| Mean dist. (km) | $327.44 \pm 100.31$ | $122.83 \pm 126.10$ | $136.26 \pm 127.17$ |
| Mean time (h) | $11.95 \pm 8.73$ | $7.23 \pm 6.88$ | $7.69 \pm 5.21$ |

## 3.2 TROPOMI versus ACE-FTS

To assess how TROPOMI's CO measurements compare with retrieved ACE-FTS profiles, we first compare these datasets on a
global scale. During the period of interest from 10 November 2017 to 31 May 2020, there were 6136 unique collocations after filtering and averaging (i.e., TROPOMI averages collocated with 6136 unique ACE-FTS observations). These collocations spanned a latitude range encompassing the polar, mid-latitude and equatorial regions, providing a broad basis for an inter-comparison of the two instruments. For the collocated observations, the mean number of TROPOMI pixels included in the averages was 11 460, indicating that the computed TROPOMI averages are statistically robust, and that pixel-to-pixel variability
or biases should be negligible. Given that each ACE-FTS solar occultation provides a CO VMR profile (typically in the altitude range of 10-150km) instead of a total column value some additional steps are needed to allow for a direct comparison between these two instruments.

As previously mentioned in Sect. 2.1, the TROPOMI CO retrieval employs a profile-scaling approach, and a single scaling factor, which represents the ratio of the retrieved to the prior column, is applied to the reference profile to obtain the "retrieved"
profile. However, these scaling factors are not provided in the TROPOMI CO product files, so these must be calculated. First, however, the CO reference profiles must be converted to partial columns, and then summed to obtain the total column concentration $c$ using the following equation:

$$c = \sum_{i=1}^{N} \frac{\Delta p_i x_i}{M_{\mathrm{da}} \bar{g}}, \tag{1}$$

where $N = 25$ is the number of layers in the TM5 *a priori* grid, $\Delta p$ is the thickness of a given partial column layer in Pa, $x$ is the
mean VMR in the layer above level $i$, $M_{da} = 0.02897$ is the molar mass of dry air in $\mathrm{kg\ mol}^{-1}$, and $\bar{g}$ is the column-averaged gravitational acceleration. The scaling factors for each collocation are then calculated by taking the ratio of the retrieved to the *a priori* total column. The scaling factor is then applied to the TM5 reference profile to obtain the "retrieved" profile, allowing for a direct comparison against ACE-FTS profiles.





Following a similar approach to what was done for the TROPOMI reference profiles, since the ACE-FTS profiles are reported
in VMR units, these must converted to partial columns as well. In addition to the VMR profiles, the ACE-FTS L2 product
includes retrieved profiles of temperature and pressure that can be used in accurately determining the partial column profile $\boldsymbol{\rho}$.
Following the method of Holl et al. (2016), this is done using the ideal gas law (Clapeyron, 1834):

$$\boldsymbol{\rho} = \frac{\boldsymbol{x}\boldsymbol{p}}{k\boldsymbol{T}}\Delta h, \tag{2}$$

where $\boldsymbol{x}$ is the VMR profile, $\boldsymbol{p}$ is the retrieved pressure in $\mathrm{Pa}$, $\boldsymbol{T}$ is the retrieved ACE-FTS temperature profile in K, $k =$
$1.380653 \times 10^{-23}$ J K$^{-1}$ is Boltzmann's constant, and $\Delta h$ is the thickness of each layer in $\mathrm{m}$. The resulting partial column
profiles only extend to the lowest ACE-FTS VMR measurement altitude, so for altitudes below this point, the partial column
profile is filled using the nearest value from the TM5 reference profile, yielding a complete partial column profile from the
surface to the top of the atmosphere (TOA).

Since ACE-FTS has a significantly higher vertical resolution than TROPOMI, the partial column profiles are linearly in-
terpolated from the 1-km altitude grid of ACE-FTS, to the 50-layer retrieval grid used by the TROPOMI SICOR retrievals.
As discussed in Sect. 2.1, for cloudy observations, TROPOMI retrievals are more sensitive to the above-cloud column than
the below-cloud portion, which is reflected in the column averaging kernel values. As a result, to correctly inter-compare the
measurements from ACE-FTS and TROPOMI, we must smooth the interpolated ACE-FTS partial column profiles with the
TROPOMI column averaging kernels. Following the methods of Borsdorff et al. (2014), Landgraf et al. (2016), and Landgraf
et al. (2018) the smoothed ACE-FTS partial column profile $\boldsymbol{\rho}^{\mathrm{smooth}}$ is given by:

$$\boldsymbol{\rho}^{\mathrm{smooth}} = \boldsymbol{A}_{\mathrm{col}}\boldsymbol{\rho}^{\mathrm{true}}, \tag{3}$$

where $\boldsymbol{A}_{\mathrm{col}}$ is the TROPOMI column averaging kernel, and $\boldsymbol{\rho}^{\mathrm{true}}$ is the ACE-FTS partial column profile interpolated to the
TROPOMI 50-layer retrieval grid. Generally, in comparisons such as this, the *a priori* profile of the higher-vertical-resolution
instrument would typically be substituted with that of the lower-vertical-resolution instrument to reduce the smoothing error
(Rodgers and Connor, 2003). However, since ACE-FTS performs solar occultation measurements, a sensitivity (i.e., the ratio
of information coming from the measurement versus the *a priori*) of 1 is assumed at each level with a negligible influence
from the *a priori* profile except at the uppermost altitudes of the ACE-FTS grid, which is beyond the ACE-FTS retrieval and
the range of the TROPOMI CO retrieval grid (which typically spans 0-50 $\mathrm{km}$) (Boone et al., 2005). As a result, a full *a priori*
substitution is not performed in the comparison of ACE-FTS with TROPOMI.

To minimize the influence of filling the missing lower altitudes of the ACE-FTS profile with the TROPOMI *a priori* profile,
the column from the lowest ACE-FTS altitude to the top of the TROPOMI retrieval grid is computed by integrating the
smoothed ACE-FTS partial column profile above the altitude of the lowest ACE-FTS measurement. Similarly, to estimate the
TROPOMI partial column in the same altitude range, the partial column below the lowest ACE-FTS altitude is computed by
summing the scaled TM5 reference profile from the surface to the lowest measured ACE-FTS altitude. This "below-ACE"
column is then subtracted from the retrieved TROPOMI total column, providing an estimate of the measured partial column
in the same altitude range as ACE-FTS, thus allowing a direct comparison of the two measurements. A similar method was





applied in Martínez-Alonso et al. (2020) for comparisons of TROPOMI's CO measurements with above-cloud partial columns computed from ATom-4 in-situ airplane profiles.

Furthermore, to assess the retrieval error associated with using the shape of the TROPOMI TM5 reference profiles to ap-
proximate the shape of the atmospheric CO profile below the lowest ACE-FTS measurement, we calculate the null-space error $e_n$ (also known as the smoothing error) following the method of Borsdorff et al. (2014), Wassmann et al. (2015), and Landgraf et al. (2016):

$$e_n = (\boldsymbol{I} - \boldsymbol{A}_{\mathrm{col}})\boldsymbol{\rho}^{\mathrm{true}}, \tag{4}$$

where $\boldsymbol{I}$ is the corresponding altitude integral operator (a unit vector in the case of a profile in partial column units), and $\boldsymbol{\rho}^{\mathrm{true}}$
represents the true CO profile (Wassmann et al., 2015). For retrievals performed over clear, cloudless scenes, the null-space error will be small since the column averaging kernel values are close to one at all altitude levels. For retrievals over cloudy scenes however, the magnitude of the null-space error is expected to be significantly larger. Here, we determine the relative null-space error (in %) in reference to the coincident unsmoothed ACE-FTS columns. If the reference profile accurately represents the true vertical trace gas distribution $\boldsymbol{\rho}^{\mathrm{true}}$, then we expect that $e_n$ should disappear and the column retrieved by TROPOMI
should be an estimate of the true total column (Landgraf et al., 2016). Furthermore, the direction of the relative null-space error (i.e., positive or negative) can indicate whether the TROPOMI reference profiles underestimate or overestimate the true vertical CO distribution.

Lastly, we compute the partial column bias values of TROPOMI with respect to ACE-FTS (TROPOMI − ACE), as well as the relative bias values (in %) between ACE-FTS and TROPOMI as (100 ×(TROPOMI−ACE)/ACE). Biases are computed
both globally, and within the following latitude bands: south Polar (90° S to 60° S), south Mid-latitudes (60° S to 20° S), Equatorial (20° S to 20° N), north Mid-latitudes (20° N to 60° N), and north Polar (60° N to 90° N) to investigate latitudinal trends in the differences.

## 3.3   TROPOMI versus PEARL-FTS

Ground-based instruments such as the PEARL-FTS provide context and a point of reference for instrument inter-comparisons
such as that of ACE-FTS and TROPOMI. In the following section describes the methods used to compare the TROPOMI and PEARL-FTS datasets. Since the PEARL-FTS only makes measurements during the period of polar sunlight, no collocations between these instruments occurred in 2017. The earliest collocation between TROPOMI and the PEARL-FTS dates to 3 March 2018, and the final collocation took place on 27 March 2020, after which mid-IR measurements by the PEARL-FTS were halted due to the lack of an on-site operator as a result of the current COVID-19 pandemic.
Similar to the methodology applied to the ACE-FTS and TROPOMI comparison, for each PEARL-FTS observation the arithmetic mean of all collocated TROPOMI pixels within a 500 km radius of Eureka is computed to reduce the pixel-to-pixel variability and enhance the statistical robustness of the comparisons. However, unlike in the ACE-FTS and TROPOMI comparison, *a priori* information is provided for both the PEARL-FTS and TROPOMI, so we perform an additional step of prior substitution to place both retrievals on a common *a priori* (in this case, the TROPOMI *a priori*) (Rodgers and Connor,





2003). This additional step serves to minimize the smoothing error when comparing two remote sensing retrievals, and a similar
method was applied for the recent comparisons of ground-based TCCON and NDACC measurements in Zhou et al. (2019), and
of TROPOMI and MOPITT by Martínez-Alonso et al. (2020). Following Rodgers and Connor (2003), the prior substitution to
obtain the optimized retrieved profile $x_{op}^{fts}$ is done by the following:

$$x_{op}^{fts} = (I - A)(x_a^{s5p} - x_a^{fts}),$$  (5)

where $I$ is the identity matrix, $A$ is the VMR averaging kernel of the PEARL-FTS, $x_a^{s5p}$ is the TROPOMI *a priori* which has
been interpolated to the PEARL-FTS retrieval grid, and $x_a^{fts}$ is the PEARL-FTS *a priori* profile.

With the PEARL-FTS VMR profile optimized with respect to TROPOMI and its *a priori* profile, the former can be inter-
polated to the TROPOMI 50-layer retrieval grid and the partial column profile calculated using the right-hand portion of Eq.
1 and the TROPOMI pressure grid. The 'best estimate' of the PEARL-FTS total column $\hat{c}^{fts}$ is determined by smoothing the

partial column profile by the TROPOMI column averaging kernel following the method of Rodgers and Connor (2003):

$$\hat{c}_{smooth}^{fts} = c_a^{s5p} + A_{col}(\rho_{op}^{fts} - \rho_a^{s5p}),$$  (6)

where $c_a^{s5p}$ is the TROPOMI *a priori* total column, $A_{col}$ is the TROPOMI column averaging kernel, $\rho_{op}^{fts}$ is the optimized
PEARL-FTS partial column profile interpolated to the TROPOMI retrieval grid, and $\rho_a^{s5p}$ is the TROPOMI *a priori* partial
column profile. In theory, this operation can be done in the opposite direction by bringing the scaled TROPOMI profile to the

PEARL-FTS retrieval grid, to then be smoothed by the PEARL-FTS averaging kernel. However, these two approaches are not
symmetrical, and one way is expected to produce a better result than the other. This is because the higher resolution will more
realistically reproduce the lower resolution measurement, rather than the other way around (Rodgers and Connor, 2003). Since
TROPOMI is the lower vertical resolution measurement in this particular instance, we chose to bring the PEARL-FTS profiles
to the TROPOMI retrieval grid.

Once the best estimate of the PEARL-FTS column with respect to TROPOMI is obtained, the bias in the retrieved TROPOMI
total columns relative to the PEARL-FTS is computed in the same manner as was done for the ACE-FTS and TROPOMI
comparison described in Sect. 3.2.

### 3.4  ACE-FTS versus PEARL-FTS

As discussed in Sect. 2.3, earlier versions of the ACE-FTS CO data product have been validated against the PEARL-FTS and

other ground-based FTSs in NDACC, namely by Clerbaux et al. (2008) and Griffin et al. (2017). Both of these studies showed
generally good agreement between ACE-FTS and the ground-based instruments. Since ACE-FTS profiles do not extend to the
surface, these previous studies primarily focused on comparisons of partial column abundances instead of total columns. In
this work, we employ a similar approach, which is described below.

Firstly, since we aim to compare the partial column abundances of ACE-FTS and the PEARL-FTS, we must determine the

optimal altitude range for the PEARL-FTS in which to perform this comparison. This step is crucial because if the selected
range is too wide, then *a priori* information may dominate the partial column comparisons, and the true vertical information



coming from the PEARL-FTS may be limited. On the other hand, if the selected altitude range is too small, then the comparison will essentially be reduced to a single layer. To achieve this, the sensitivity of the retrievals at each level $k$ was first computed by summing the corresponding rows of the averaging kernel matrix, $\sum_i \boldsymbol{A}_{ki}$, following Vigouroux et al. (2008). The sensitivity

density (i.e., the fraction of retrievals with sensitivity at a given altitude) of the PEARL-FTS retrievals was then investigated for all collocated ACE-FTS measurements (Fig. 1). From an analysis of the sensitivity density, an optimal altitude range of $10.25 - 40.17$ km was selected for the comparison of the partial columns. This chosen range is similar to the the altitude range of $9.0 - 48.5$ km used by Griffin et al. (2017), albeit slightly more conservative. However, the SFIT4 CO retrieval has been modified in the meantime due to an NDACC-wide harmonization initiative, and the range used by Griffin et al. (2017) may

no longer be ideal. A smaller altitude region with high sensitivity can be seen between $0.61 - 2.21$ km, however ACE-FTS retrieved profiles do not typically extend this close to the surface, and as a result this region was not used.

Again drawing from Rodgers and Connor (2003), since the PEARL-FTS is of a lower vertical resolution than ACE-FTS the retrieved ACE VMR profiles must be interpolated to the coarser altitude grid of the PEARL-FTS. However, since the retrieval grid of the PEARL-FTS ($0.61$ km to $120$ km) extends further towards the surface than ACE-FTS, the bottom-most altitudes of

each coincident ACE-FTS VMR profile beneath the lowest measurement must first be filled in using the nearest value from the PEARL-FTS *a priori* profile. In this case, since it is assumed that ACE-FTS has a sensitivity of 1 at each measured altitude, and no *a priori* information is provided with the ACE data, we do not perform any prior substitution step here. ACE-FTS VMR profiles are then smoothed using the VMR averaging kernel $\boldsymbol{A}$ of the PEARL-FTS using a similar form to Eq. 6 (Rodgers and Connor, 2003):

$$\boldsymbol{x}^{\mathrm{ace}}_{\mathrm{smooth}} = \boldsymbol{x}^{\mathrm{fts}}_a + \boldsymbol{A}(\boldsymbol{x}^{\mathrm{ace}} - \boldsymbol{x}^{\mathrm{fts}}_a), \tag{7}$$

where $\boldsymbol{x}^{\mathrm{ace}}_{\mathrm{smooth}}$ is the smoothed ACE-FTS VMR profile, $\boldsymbol{x}^{\mathrm{fts}}_a$ is the PEARL-FTS *a priori*, and $\boldsymbol{x}^{\mathrm{ace}}$ is the original ACE-FTS profile that has been interpolated to the PEARL-FTS retrieval grid. The partial column profile for ACE-FTS is calculated using Eq. 2, and then the partial columns between $10.25 - 40.17$ km are summed. The difference between the ACE-FTS and the PEARL-FTS partial columns, $\delta c_{\mathrm{pc}}$, is found by:

$$\delta c_{\mathrm{pc}} = c^{\mathrm{ace}}_{\mathrm{pc}} - c^{\mathrm{fts}}_{\mathrm{pc}}, \tag{8}$$

where $c^{\mathrm{ace}}_{\mathrm{pc}}$ and $c^{\mathrm{fts}}_{\mathrm{pc}}$ are the ACE-FTS and PEARL-FTS partial columns respectively, between $10.25 - 40.17$ km.



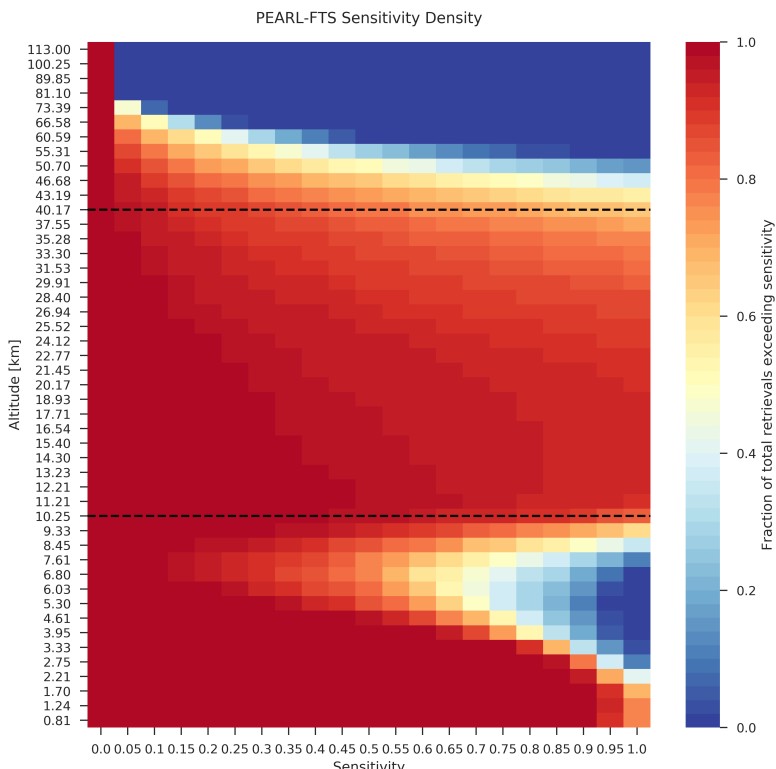

**Figure 1.** The mean sensitivity density of the PEARL-FTS CO retrieval for all collocated ACE-FTS measurements. The y-axis altitudes correspond to the lower boundaries of the PEARL-FTS retrieval layers, and the black dashed lines denote the selected altitude range for the partial column comparisons of 10.25 to 40.17 km.

## 4   Results and discussion

### 4.1   TROPOMI versus ACE-FTS: global comparison

A global comparison of ACE-FTS and TROPOMI partial columns was performed for the period from 10 November 2017
to 31 May 2020. During this period, there were a total of 6136 unique collocations, spanning 82° N to 81° S and broadly
encompassing all longitudinal meridians. Due to the nature of the overlap between the ACE-FTS and TROPOMI orbits, a
higher density of collocations occurred at the higher latitudes (both north and south) than towards the equator.

Linear regressions were performed and the mean biases and standard deviations were computed for the global comparison,
as well as in five latitude bands; the north Polar region (60° N to 90° N), northern Mid-latitudes (20° N to 60° N), the Equatorial



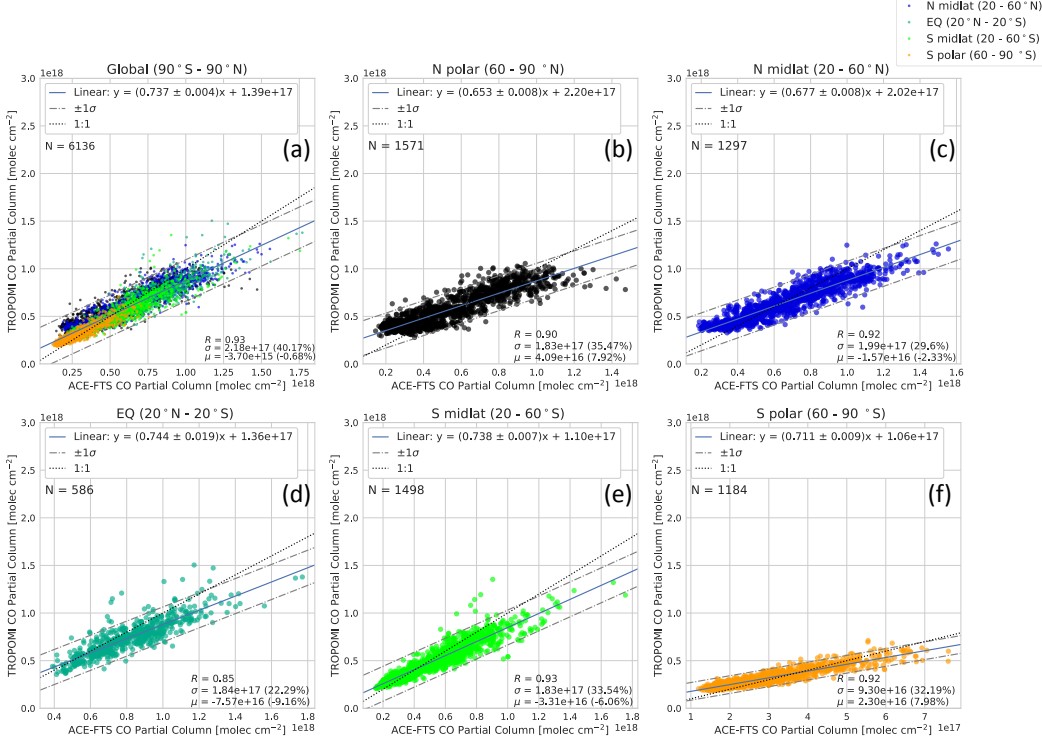

**Figure 2.** Correlation plots of collocated ACE-FTS and TROPOMI partial columns in the following latitude bands: **(a)** Global (90° S to 90° N), **(b)** N Polar (60° N to 90° N), **(c)** northern Mid-latitudes (20° N to 60° N), **(d)** Equatorial (20° S to 20° N), **(e)** southern Mid-latitudes (20° S to 60° S), and **(d)** S Polar (60° S to 90° S). In panel **(a)**, the color of the data points corresponds to the respective latitude regions. Values of the Pearson correlation coefficient $R$, the standard deviation of the TROPOMI columns $\sigma$, and the mean bias $\mu$ of the respective latitude band are displayed in the lower right of each panel.

region (20° S to 20° N), southern Mid-latitudes (60° S to 20° S), and the south Polar region (90° S to 60° S). Figure 2 and Table 3 show the results of these comparisons. Globally, there is very strong correlation between the measurements from both instruments ($R = 0.93$), with a small mean bias of $-3.70 \times 10^{15} \pm 1.37 \times 10^{15}$ molec. cm$^{-2}$ ($-0.68 \pm 0.25$ %; bias ± SEM), and a standard deviation of the differences of $1.07 \times 10^{17}$ molec. cm$^{-2}$ (19.79 %). The observed global mean bias between ACE-FTS and TROPOMI is well within the mission accuracy requirement of ± 15 % (Landgraf et al., 2016), and is consistent

with global comparisons of the CO product to the ECMWF Integrated Forecasting System (IFS) by Borsdorff et al. (2018a) who found a global mean relative bias of $3.2 \pm 5.5$ % (bias ± standard deviation).

From the latitudinal comparisons between ACE-FTS and TROPOMI shown in Fig. 2 and summarized in Table 3, it can be seen that the magnitude (as well as the sign) of the biases varies by latitude band. The largest positive relative biases are observed in the north and south Polar regions, with mean differences of $4.09 \times 10^{16} \pm 2.99 \times 10^{15}$ molec. cm$^{-2}$ and

$2.30 \times 10^{16} \pm 1.46 \times 10^{15}$ molec. cm$^{-2}$ ($7.92 \pm 0.58$ % and $7.98 \pm 0.51$ %) respectively, indicative of high TROPOMI





**Table 3.** Summary of the number of collocations, the mean partial column differences, and the standard deviations of the differences between ACE-FTS and TROPOMI globally, and in each latitude region. The relative bias and standard deviation values are computed with respect to ACE-FTS (i.e., $100\times$(TROPOMI$-$ACE-FTS)/ACE-FTS). The uncertainties provided for the absolute and relative biases corresponds to the standard error of the mean.

| Region (latitude) | $N_{\text{collocations}}$ | $R$ | Mean abs. bias (molec. cm$^{-2}$) | $\sigma_{bias}$ (molec. cm$^{-2}$) | Mean rel. bias (%) | Rel. $\sigma_{bias}$ (%) |
|---|---|---|---|---|---|---|
| Global (90° S to 90° N) | 6136 | 0.93 | $-3.70\times10^{15}\ \pm1.37\times10^{15}$ | $1.07\times10^{17}$ | $-0.68\ \pm0.25\%$ | 19.79 % |
| N Polar (60° N to 90° N) | 1571 | 0.90 | $4.09\times10^{16}\ \pm2.99\times10^{15}$ | $1.19\times10^{17}$ | $7.92\ \pm0.58\%$ | 22.97 % |
| N Mid-lat (20° N to 60° N) | 1297 | 0.92 | $-1.57\times10^{16}\ \pm3.26\times10^{15}$ | $1.17\times10^{17}$ | $-2.33\ \pm0.49\%$ | 17.47 % |
| Equator (20° S to 20° N) | 586 | 0.85 | $-7.57\times10^{16}\ \pm4.54\times10^{15}$ | $1.10\times10^{17}$ | $-9.16\ \pm0.55\%$ | 13.31 % |
| S Mid-lat (20° S to 60° S) | 1498 | 0.93 | $-3.31\times10^{16}\ \pm2.32\times10^{15}$ | $8.98\times10^{16}$ | $-6.06\ \pm0.42\%$ | 16.45 % |
| S Polar (60° S to 90° S) | 1184 | 0.92 | $2.30\times10^{16}\ \pm1.46\times10^{15}$ | $5.04\times10^{16}$ | $7.98\ \pm0.51\%$ | 17.45 % |

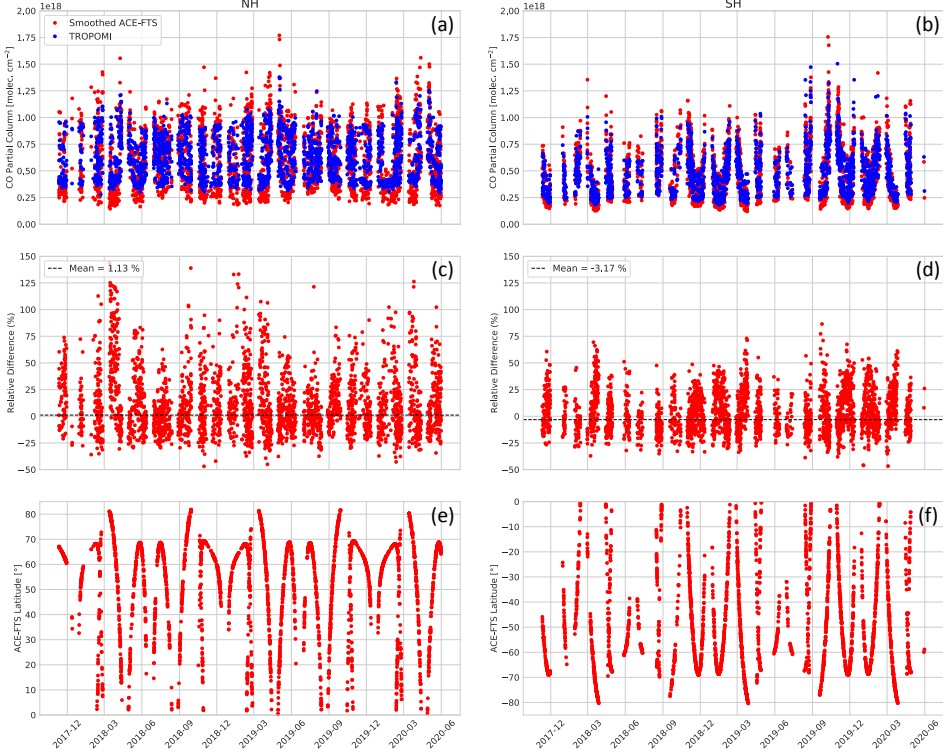

**Figure 3.** **(a)** and **(b)** Time-series of smoothed ACE-FTS and TROPOMI partial columns, **(c)** and **(d)** the relative differences between the instruments, and **(e)** and **(f)** the latitude of the coincident ACE-FTS measurement for both the northern hemisphere (left column) and the southern hemisphere (right column). The black dashed horizontal lines in the middle panels denote the mean of the differences.



column values in the polar regions relative to ACE-FTS. The largest negative relative bias was found in the Equatorial region, with a mean difference of $-7.57 \times 10^{16} \pm 4.54 \times 10^{15}$ molec. cm$^{-2}$ ($-9.16 \pm 0.55$ %). Smaller negative biases of $-1.57 \times 10^{16} \pm 3.26 \times 10^{15}$ molec. cm$^{-2}$ ($-2.33 \pm 0.49$ %) and $-3.31 \times 10^{16} \pm 2.32 \times 10^{15}$ molec. cm$^{-2}$ ($-6.06 \pm 0.42$ %) are seen in the northern and southern Mid-latitude regions respectively. The standard deviations of the mean relative differ-
ences range between 13.31 % (Equatorial region) and 22.97 % (north Polar region). Despite the variability in the mean of the differences with respect to latitude, generally strong correlations between ACE-FTS and TROPOMI are observed across all latitude bands, with the weakest correlation occurring in the Equatorial region ($R = 0.85$), which may be due in part to the smaller overall number of collocations ($N = 586$) in this latitude band relative to all others.

Overall, these observed correlations suggest that both instruments measure similar temporal trends in CO partial columns
globally. Time-series of the TROPOMI and smoothed ACE-FTS partial columns, their relative differences, and the latitude of the coincident ACE-FTS measurements are shown for both the northern and southern hemispheres (NH & SH) in Fig. 3. From this, it is clear that both instruments do observe similar seasonal cycles in the CO columns, particularly in the SH

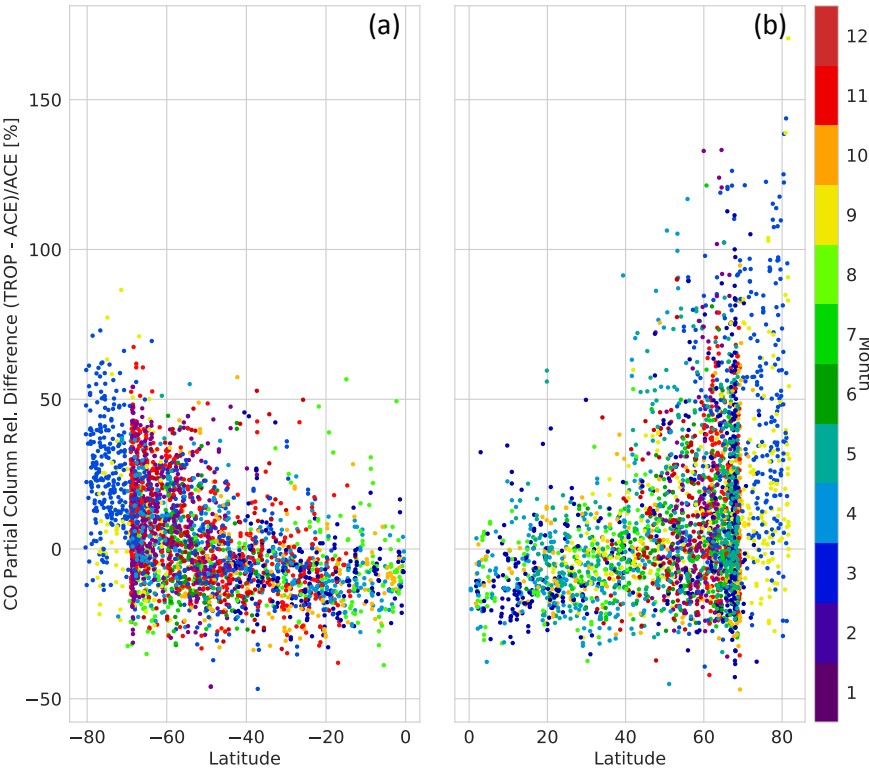

**Figure 4.** Relative difference between TROPOMI and ACE-FTS versus latitude in **(a)** the southern hemisphere and **(b)** the northern hemisphere for the period from 10 November 2017 to 31 May 2020. The data points are binned by color depending on the month in which the collocation occurred.





where anthropogenic CO sources are less influential, and overall no clear seasonal dependence of the biases is apparent. The aforementioned latitudinal variability in the biases, however, can still be observed in panels **(c)** and **(d)** of Fig. 3. The largest

relative differences between the two instruments can be seen during March and September of each year when collocations are occurring at high latitudes in both hemispheres (i.e., towards the polar regions), while generally smaller relative differences, conversely, are observed for collocations occurring at lower latitudes (i.e., nearer to the equator). It can also be noted that the dynamic range of ACE-FTS partial column values is noticeably larger than TROPOMI in both hemispheres.

To examine the relationship between the partial column differences and latitude, the differences versus the latitude of each

collocation are shown in Fig. 4. On average, larger differences occur at the higher latitudes (most notably in the northern hemisphere), with smaller or negative differences present towards the equator. A similar pattern in the biases of the TROPOMI CO product was observed in comparisons with the ECMWF-IFS model in Borsdorff et al. (2018a), which displayed negative biases near the equator, and higher positive biases on the order of 10 % towards the poles. Comparisons of the TROPOMI CO product to ATom-4 in-situ aircraft profiles in Martínez-Alonso et al. (2020) displayed no latitudinal dependence in the biases,

however, these comparisons were limited to only 103 collocated profiles over a smaller geographical and latitudinal range (60° S to 85° N).

To assess whether any differences are introduced by the TROPOMI retrievals over cloudless versus cloud-covered scenes, the mean differences between ACE-FTS and TROPOMI were independently investigated for clear-sky and cloudy scenes (in addition to all scenes), and are shown in Fig. 5 for both the unsmoothed and smoothed cases. In general, smoothing

ACE-FTS by the TROPOMI column averaging kernels reduces the mean relative bias by a significant margin both in the global comparison as well as in all distinct latitude bands, but yields slightly poorer correlations in some regions (maximum difference of 0.03 in the Pearson correlation coefficients). The smoothing operation has a noticeably larger effect in the cloud-covered scenes than for the clear-sky pixels, and it shifts the mean biases in the Equatorial and mid-latitude regions from positive to slightly negative. Furthermore, in both the unsmoothed and smoothed cases, the clear-sky-only scenes tend to be

biased higher than the clear+cloudy scenes and cloud-covered scenes only. It should also be noted that particularly in the unsmoothed case, there is consistently better correlation between ACE-FTS and TROPOMI for cloud-covered vs. clear-sky scenes. This observed tendency is related to the aforementioned changes in the shape of the TROPOMI column averaging kernels over clear versus cloudy scenes. As outlined in Sect. 2.1, the shape of the TROPOMI column averaging kernels varies based on the cloud fraction of the measurement to reflect a reduction in sensitivity of the retrieval due to cloud contamination.

For observations over clear-sky scenes, the values of the column averaging kernel will be close to one at all levels, and the influence of the reference profile on the computation of the scaling factor will be minimal. However, for measurements made over cloud-covered scenes, the column averaging kernel values rapidly decrease towards zero below the height of the cloud, while simultaneously increasing above the cloud. Because of this, in cloudy scenes, the above-cloud column (which is in the same approximate altitude range that ACE-FTS measures) is used to estimate the total column, and a greater reliance is placed

on the TM5 reference profiles. If the reference profiles are underestimating the CO column below the height of the cloud, then the resulting retrieved total column value will be biased lower, which is broadly consistent with the observed relationship.



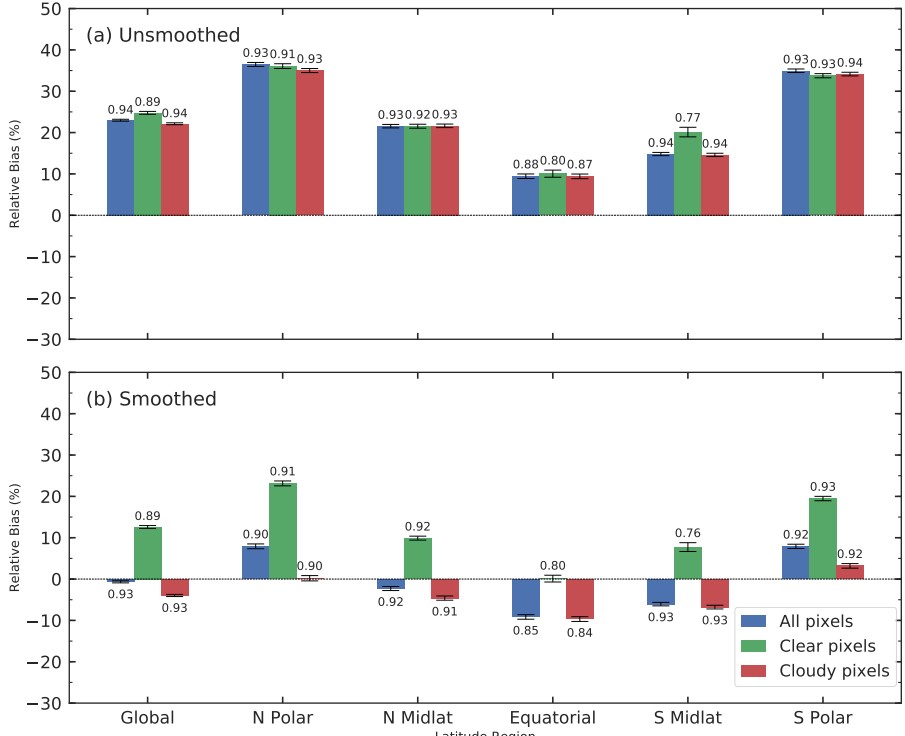

**Figure 5.** Summary of the relative differences between TROPOMI and ACE-FTS for **(a)** the unsmoothed and **(b)** smoothed comparisons for all TROPOMI pixels (qa_value $\geq$ 0.5; blue bars), clear pixels only (qa_value = 1.0; green bars), and cloud-covered pixels only (0.5 $\leq$ qa_value $\leq$ 0.7; red bars). The error bars correspond to the standard errors and the values above/below the error bars are the Pearson correlation coefficients for that particular case and latitude region.

Despite the differences between the unsmoothed and smoothed comparisons, both cases still display a latitudinal bias, with the largest mean differences occurring in the NH and SH polar regions.

As discussed in Sect. 3.2, the null-space error $e_n$ can be helpful in diagnosing the error associated with the choice of the

*a priori* profile shape on the retrieved CO column in a profile-scaling approach, particularly for measurements made over cloudy scenes. The null-space error was computed for all collocated cloudy pixels (0.5 $\leq$ qa_value $\leq$ 0.7) relative to the true (unsmoothed) ACE-FTS profiles, as shown in Fig. 6. The values of the relative null-space error are entirely negative across all latitudes, with a global mean of $-28.58 \pm 9.98$ % (bias $\pm$ standard deviation). The negative nature of the null-space errors suggests that the TM5 references profiles are generally underestimating CO concentrations in the vertical extent with respect

to the retrieved ACE-FTS columns. Furthermore, a pattern in the relative null-space error with respect to latitude can also be observed, with the most strongly negative values occurring between 60° to 90° in both the NH and SH. The larger observed difference in this latitude band may be due to differing cloud properties relative to lower latitudes regions, such as cloud height and optical thickness. This implies that the magnitude of the error associated with this choice of reference profile is on average



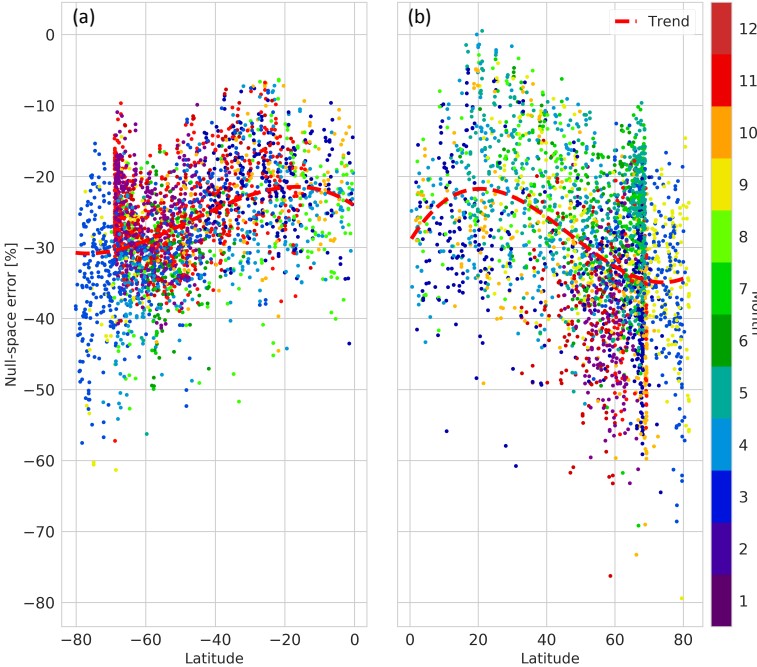

**Figure 6.** The relative null-space error of TROPOMI pixels over cloudy scenes in **(a)** the southern hemisphere and **(b)** the northern hemisphere with respect to the true (unsmoothed) ACE-FTS partial columns versus latitude for the period from 10 November 2017 to 31 May 2020. A 3rd-order polynomial fit (denoted by the dashed red line) was applied to the data to better highlight the underlying pattern.

larger in these high latitude regions. In general, for collocations where the null-space error is more strongly negative (i.e.,
the below-cloud column is more largely underestimated), a larger positive bias in the TROPOMI partial column relative to
ACE-FTS is anticipated.

The correlation between the relative differences and the relative null-space errors was also investigated in the same latitude
bands as the partial column comparisons, and this is shown in Fig. 7. In the upper left panel of Fig. 7, no clear relationship
between the relative null-space errors ($R = 0.04$) can be seen in the global comparison. However, within the latitude bands,
weak correlations between the null-space error and the partial column differences can be observed. In particular, in the N Polar,
Equatorial, and S Polar regions, the relative partial column differences increase with relative null-space errors, with Pearson
correlation coefficients of $R = 0.22$, $R = 0.41$, and $R = 0.20$, respectively. The north and south Polar regions display the
most strongly negative mean relative null-space errors, with $-33.18 \pm 10.68$ % and $-29.36 \pm 7.92$ %, respectively, while the
Equatorial region has the least negative mean null-space error with $-23.15 \pm 7.98$ %. In the northern and southern Mid-latitude
regions, no notable correlation between the relative null-space error and the relative partial column differences is observed, with
$R = 0.10$ and $R = -0.11$, respectively. Overall, the observed pattern in the mean relative null-space errors suggests that relative





to retrieved ACE-FTS columns, the error associated with the choice of the TM5 reference profiles is largest in the N and S Polar regions, and lowest in the Equatorial region.

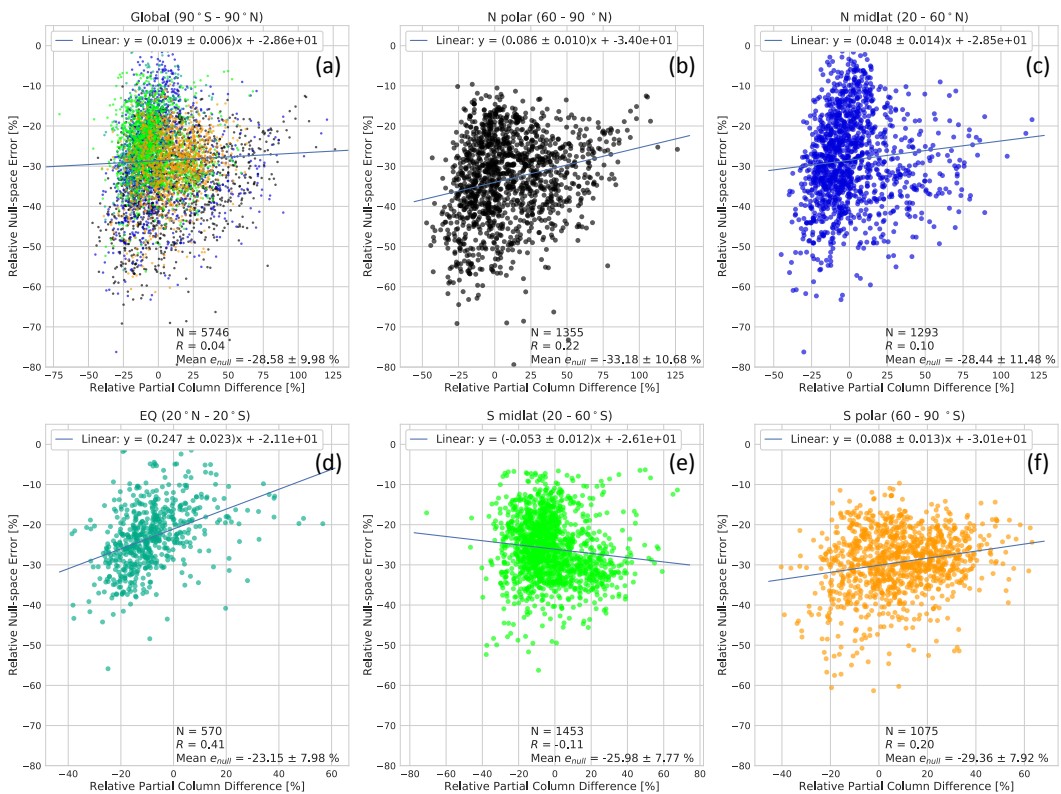

**Figure 7.** Correlation plots of the relative null-space error of the TROPOMI retrievals versus the relative partial column differences between ACE-FTS and TROPOMI in the same latitude bins as Fig. 2. See the caption of Fig. 2 for more details.

## 4.2 High-Arctic ground-based comparisons

### 4.2.1 TROPOMI versus PEARL-FTS

A total of 1875 collocations between TROPOMI and the PEARL-FTS at Eureka, Nunavut were found from 3 March 2018 to 27 March 2020. Correlation plots of TROPOMI total columns versus both the unsmoothed and smoothed PEARL-FTS total columns are displayed in the left and right panels of Fig. 8, respectively. Smoothing the PEARL-FTS profiles by the TROPOMI column averaging kernels has a significant effect on the agreement between the two instruments. In the unsmoothed

comparison, a correlation is observed between the two instruments ($R = 0.84$), but the slope of the linear fit is 1.75 and there is a large mean positive bias of $8.89 \times 10^{17} \pm 3.93 \times 10^{15}$ molec. cm$^{-2}$ ($73.7 \pm 0.33$ %) with a standard deviation of the



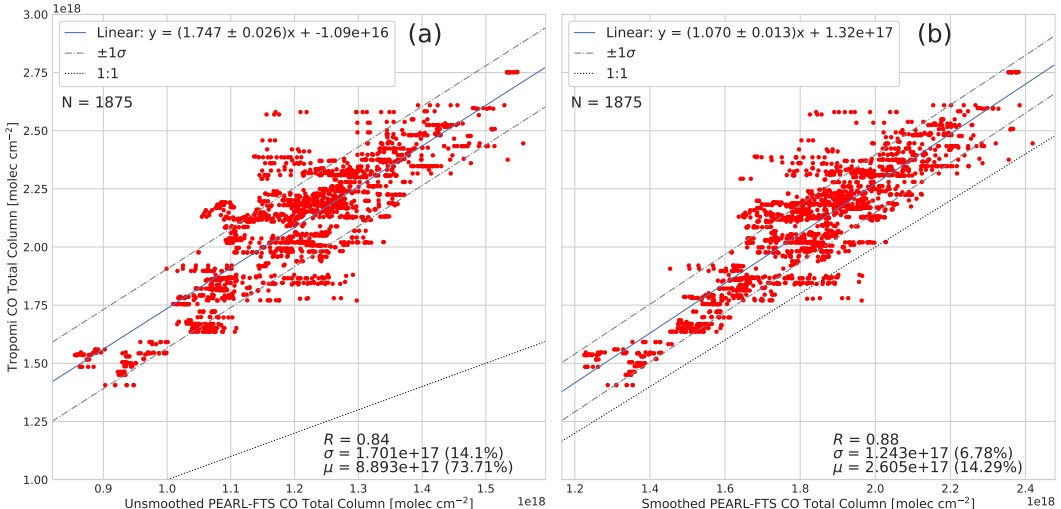

**Figure 8.** Correlation plots of TROPOMI CO total columns with **(a)** unsmoothed and **(b)** smoothed PEARL-FTS CO columns.

differences of $1.70 \times 10^{17}$ molec. $\mathrm{cm}^{-2}$ (14.1 %). The correlation with smoothed PEARL-FTS columns is improved ($R = 0.88$, slope of linear fit = 1.07), and the mean bias was reduced to $2.61 \times 10^{17} \pm 2.87 \times 10^{15}$ molec. $\mathrm{cm}^{-2}$ (14.3 ± 0.16 %), with a standard deviation of $1.243 \times 10^{17}$ molec. $\mathrm{cm}^{-2}$ (6.78 %). While smoothing the PEARL-FTS retrievals by the TROPOMI

column averaging kernels reduced the mean bias by approximately 60 %, a systematic bias of 14.3 % is still present. The observed positive mean bias is consistent with the recent ground-based validation efforts of Sha et al. (2021), who found a bias of 12.96 ± 4.56 % for TROPOMI versus the PEARL-FTS while using a stricter collocation criterion of 50 km in space and 3 hours in time, and is also generally consistent with the positive biases observed between ACE-FTS and TROPOMI over the north Polar region.

To ascertain whether there is a seasonal dependence in the biases between TROPOMI and the PEARL-FTS, the time-series of the TROPOMI and smoothed PEARL-FTS total columns is shown in the top panel of Fig. 9, along with the total column and relative differences. From Fig. 9, it can be seen that with the exception of a few collocations during the late spring and early summer of 2018 and 2019, a positive systematic bias is present in the TROPOMI CO total columns with respect to the smoothed PEARL-FTS CO total columns. Furthermore, the differences display some seasonal variability, with the largest

differences typically present during the spring, and the lowest differences occurring in the summer months. The larger CO column biases in the early spring may be a result of polar vortex conditions accompanied by the descent of mesospheric air-masses containing high concentrations of CO as the vortex begins to dissipate, an event previously observed over Eureka in Manney et al. (2008). Furthermore, larger differences may arise during the spring months from a mismatch in the TROPOMI footprint and the broader spatial extent of the PEARL-FTS measurements at high solar zenith angles (i.e., the slant-path of

the PEARL-FTS covers a greater horizontal distance in high SZA conditions). In general, both instruments capture the same

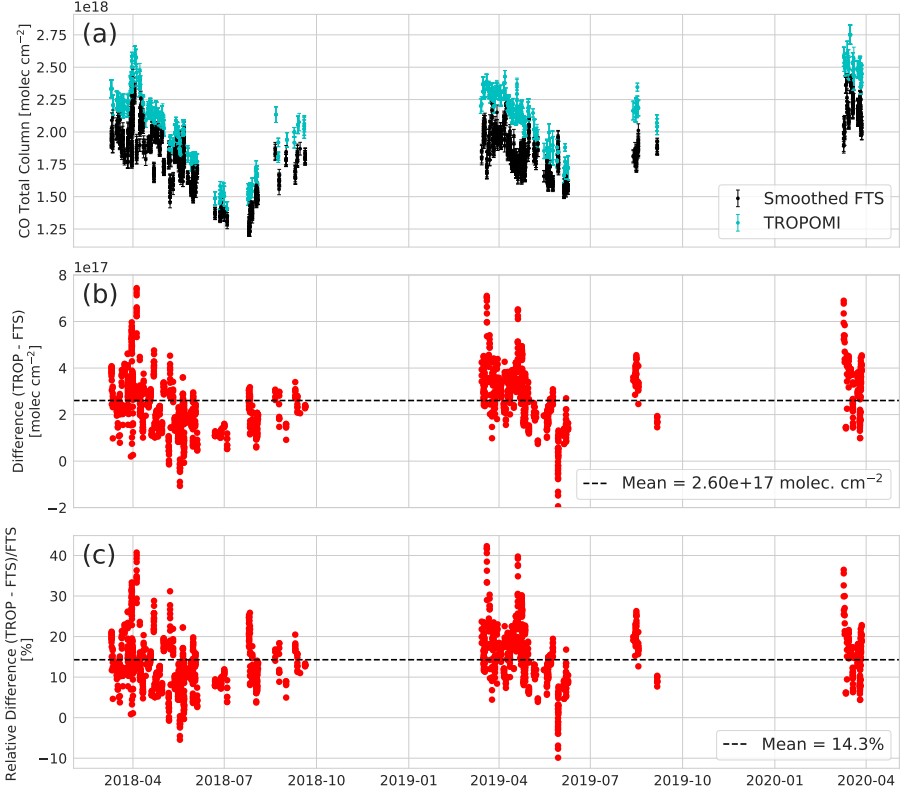

**Figure 9.** Time-series of **(a)** TROPOMI and smoothed PEARL-FTS total columns, **(b)** absolute column differences (in molec. $cm^{-2}$), and **(c)** relative differences (in %). The error bars in the top panel represent the measurement uncertainties of both the PEARL-FTS and TROPOMI.

temporal patterns in the CO total columns across all months for which comparisons were possible, however TROPOMI displays a consistent systematic high bias in the high-Arctic region within 500 km of Eureka.

### 4.2.2 ACE-FTS versus the PEARL-FTS

Comparison of ACE-FTS and PEARL-FTS CO partial columns provides additional context for the TROPOMI results presented
above. Here, a total of 2906 unique collocations between ACE-FTS and the PEARL-FTS were analyzed spanning the period from 25 February 2007 to 18 March 2020. As outlined in Sect. 3.4, partial columns in the altitude range of $10.25 - 40.17$ km are compared. The vertical information content of the PEARL-FTS is given by the degrees of freedom for signal (DOFS) which is calculated from the trace of the averaging kernels. The collocated PEARL-FTS retrievals have a mean total column DOFS of $2.2 \pm 0.37$, while in the range of 10.25 to 40.17 km the mean DOFS is $0.55 \pm 0.24$. A DOFS of 1 or greater in
the selected altitude range would be ideal, however a DOFS of 0.55 implies that there is approximately a quarter of the total vertical information coming from the measurement this range.





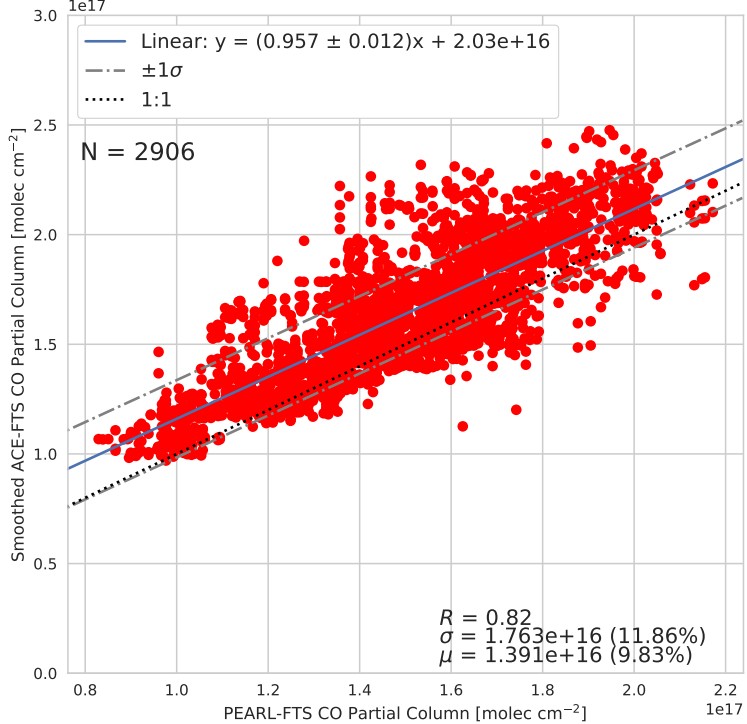

**Figure 10.** Correlation plot for ACE-FTS vs. PEARL-FTS CO partial columns in the range of 10.25 to 40.17 km for the period from 25 February 2007 to 18 March 2020. $R$, $\sigma$, and $\mu$ are defined in the same way here as in Fig. 2.

The correlation plot for ACE-FTS and PEARL-FTS partial columns in the range of $10.25 - 40.17$ km is shown in Fig. 10. Good agreement is observed between ACE-FTS and the PEARL-FTS ($R = 0.82$, slope of linear fit = 0.96), with a mean difference of $1.39{\times}10^{16} \pm 3.27{\times}10^{14}$ molec. cm$^{-2}$ ($9.83 \pm 0.22$ %; bias $\pm$ SEM) with a standard deviation of $1.76{\times}10^{16}$ molec. cm$^{-2}$ (11.86 %). This observed relative bias is similar to the findings of Griffin et al. (2017), who obtained a mean relative difference of $7.1 \pm 1.8$ % with a correlation of $R = 0.80$ between ACE-FTS and the PEARL-FTS for an earlier version of the ACE CO data product (v3.5). Although these findings do not overlap within their bounds of uncertainty on the standard error, the ACE-FTS and PEARL-FTS retrievals have each been updated in the meantime since the study by Griffin et al., which likely accounts for the small additional difference. Both TROPOMI and ACE-FTS display high systematic biases relative to the PEARL-FTS, however the observed mean relative bias in ACE-FTS relative to the PEARL-FTS is lower than for TROPOMI (9.83 % versus 14.2 %, respectively). In general, this result is consistent with the two previous comparisons performed in this work (i.e., both TROPOMI and ACE-FTS are biased high relative to the PEARL-FTS, but TROPOMI is biased higher relative the PEARL-FTS than ACE-FTS), which suggests that the observed high bias in TROPOMI over the high Arctic is a genuine feature in the TROPOMI CO product.



## 5 Conclusions

The TROPOMI instrument provides a highly spatially-resolved global data set of CO columns. However, the validity and accuracy of TROPOMI's CO product in remote regions such as the high Arctic has previously not been well characterized. In this work, we have compared TROPOMI, ACE-FTS and a high-Arctic ground-based FTS located in Eureka, Nunavut. A global comparison of TROPOMI with ACE-FTS CO partial columns was performed for the period from 10 November 2017 to 31 May 2020, resulting in excellent agreement, with a Pearson correlation coefficient of $R = 0.93$, and a mean relative bias of $-0.68 \pm 0.25$ % globally. The agreement between TROPOMI and ACE-FTS was also investigated in five latitude bands including: the north Polar region (60° N to 90° N), the northern Mid-latitudes (20° N to 60° N), the Equatorial region (20° S to 20° N), the southern Mid-latitudes (20° S to 60° S), and the south Polar region (60°S to 90° S). A latitudinal dependence on the mean differences was observed, with positive mean relative biases of $7.92 \pm 0.58$ % and $7.98 \pm 0.51$ % in the north and south Polar regions, respectively, while a negative bias of $-9.16 \pm 0.55$ % was found in the Equatorial region. This observed trend is generally consistent with earlier comparisons of the TROPOMI CO product with the ECMWF-IFS model in Borsdorff et al. (2018a). Furthermore, to highlight any differences introduced by cloud contamination in the TROPOMI CO measurements, the latitudinal comparisons were repeated independently for cloud-covered and clear scenes only, and for the unsmoothed and smoothed cases. Clear-sky pixels were found to be biased higher with slightly poorer correlations on average than clear+cloudy scenes and cloud-covered scenes only, which suggests that the TM5 reference profile shape used in the retrieval can have a measurable effect on the TROPOMI columns in the comparisons. Additionally, the latitudinal dependence of the biases is present in both the unsmoothed and smoothed cases. Despite the observed variability in the magnitude and direction of the mean biases, strong correlations ranging from $R = 0.93$ (SH mid-latitude region) to $R = 0.85$ (Equatorial region) were found between TROPOMI and ACE-FTS across all latitude bands.

To provide additional context to the global comparison of TROPOMI with ACE-FTS in the Arctic, both satellite data products were compared against NDACC measurements from the PEARL-FTS in Eureka, Nunavut (80.05° N, 86.42° W). Comparisons of TROPOMI with smoothed PEARL-FTS total columns in the period from 3 March 2018 to 27 March 2020 showed that the datasets were strongly correlated ($R = 0.88$, slope of linear fit = 1.07), however a systematic mean positive bias of $14.3 \pm 0.16$ % was also observed. These results are consistent with recent ground-based validation efforts by Sha et al. (2021) who found a comparable mean bias of $12.96 \pm 4.56$ % (bias $\pm$ standard deviation) for the PEARL-FTS while using stricter collocation criterion than in this study. A small degree of seasonal variability in the differences was found, with larger mean biases on average occurring during the spring months, and the lowest biases present during the summer months. However, with the exception of a few collocations during the late spring and early summer of 2018 and 2019, TROPOMI was consistently biased higher than the PEARL-FTS. Lastly, a partial column comparison between ACE-FTS and the PEARL-FTS was performed for the period from 25 February 2007 to 18 March 2020. These comparisons were performed in the optimal altitude range of $10.25 - 40.17$ km, which was determined from an analysis of the sensitivity density of all PEARL-FTS retrievals that were collocated with ACE-FTS measurements. These partial column comparisons showed good agreement ($R = 0.82$, slope of linear fit = 0.96), and a mean positive bias of $9.83 \pm 0.22$ % in ACE-FTS with respect to the ground-based





FTS. These findings are similar to previous validation results in Griffin et al. (2017), who found a mean relative difference of

$7.1 \pm 1.8$ % between ACE-FTS and the PEARL-FTS for an earlier version of the ACE-FTS CO data product (v3.5).

In general, the magnitude and sign of the mean relative differences are consistent across all inter-comparisons presented in this work, suggesting that the current TROPOMI CO product exhibits a high bias in the high-Arctic region that is consistent with the recent ground-based validation results of Sha et al. (2021). The observed mean differences fall within the TROPOMI mission accuracy requirement of $\pm$ 15 %, indicating that the data quality of the CO product in these high-latitude regions

meets the specifications. Proposed updates to the TROPOMI CO retrieval spectroscopy and de-striping methods described in Borsdorff et al. (2019) are expected to improve the latitudinal bias that is currently present in the operational product. It is suggested that a similar validation exercise be repeated following the release of the upcoming version 2 TROPOMI CO product to verify that the observed latitudinal bias has been corrected.

*Data availability.* TROPOMI level 2 CO retrievals and reference profiles for the period from 10 November 2017 to 1 June 2020 were

downloaded from https://s5pexp.copernicus.eu/. ACE-FTS retrievals can be accessed at https://databace.scisat.ca/level2/ace_v4.1/NETCDF/ (registration required), and the ACE-FTS data quality flags used for filtering the dataset can be accessed at https://doi.org/10.5683/SP2/ BC4ATC. PEARL-FTS CO retrievals were obtained from https://ftp.cpc.ncep.noaa.gov/ndacc/station/eureka/hdf/ftir/.

*Author contributions.* TW, KS, and KW developed the concept and methodology of the paper. TW performed the formal data analysis, software development, presentation of the results, and the PEARL-FTS CO retrievals for the period from 2018 to 2020. EL performed the

PEARL-FTS CO retrievals for the period from 2006 to 2018. TB and JL oversaw the TROPOMI data analysis. All authors discussed the results and provided feedback on the manuscript.

*Competing interests.* The authors declare that they have no conflict of interest.

*Acknowledgements.* Funding for this work was provided by the Natural Sciences and Engineering Research Council (NSERC) Canadian Climate Change and Atmospheric Research (CCAR) Probing the Atmosphere of the High Arctic (PAHA) project (grant no. 433842-2012),

and the Canadian Space Agency (CSA) Earth System Science Data Analyses (ESSDA) grant (reference no. 16SUASCMEV). PEARL-FTS measurements were made at PEARL by the Canadian Network for the Detection of Atmospheric Composition Change (CANDAC), which has been supported by the Atlantic Innovation Fund/Nova Scotia Research Innovation Trust, Canada Foundation for Innovation, Canadian Foundation for Climate and Atmospheric Sciences, the CSA, Environment and Climate Change Canada (ECCC), Government of Canada International Polar Year funding, NSERC, Northern Scientific Training Program (NSTP), Ontario Innovation Trust, Polar Continental Shelf

Program, and Ontario Research Fund. The authors thank CANDAC/PEARL/PAHA PI James Drummond, Canadian Arctic ACE/OSIRIS Validation Campaign PI Kaley A. Walker, PEARL Site Manager Pierre Fogal, CANDAC Data Manager Yan Tsehtik, the CANDAC oper-





ators, and the staff at ECCCs Eureka Weather Station for their contributions to data acquisition, and for logistical and on-site support. The Atmospheric Chemistry Experiment is a Canadian-led mission mainly supported by the Canadian Space Agency. Sentinel-5 Precursor is a European Space Agency (ESA) mission implemented on behalf of the European Commission (EC). The TROPOMI payload is a joint development by ESA and the Netherlands Space Office (NSO). The Sentinel-5 Precursor ground-segment development has been funded by ESA with national contributions from The Netherlands, Germany, and Belgium.



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
