# Peer review of "Inter-comparison of CO measurements from TROPOMI, ACE-FTS, and a high-Arctic ground-based FTS"

_Atmospheric Measurement Techniques, 2021_

## Author Comment (AC1)

**Response to reviewers**

**AMTD Manuscript doi: 10.5194/amt-2021-190**
**Title: "Inter-comparison of CO measurements from TROPOMI, ACE-FTS, and a high-Arctic ground-based FTS" by Wizenberg et al.**

We would like to thank both reviewers for their helpful and thoughtful comments and suggestions. Reviewer comments are in blue, author responses are in black, and any additions to the text are underlined. The line and page numbers provided correspond to the version of the manuscript available on AMTD.
* * *
**Responses to Reviewer #1**

**Comment C1.1 –** L50: add TES +reference

**Reply:** Reference has been added for TES: "The TROPOspheric Monitoring Instrument (TROPOMI) provides the highest spatially resolved measurements of CO from space currently available, and is extending the global CO record established by previous satellite instruments including Measurements of Pollution In The Troposphere (MOPITT; Drummond and Mand, 1996), the Tropospheric Emission Spectrometer (TES; Beer et al., 2001), the Atmospheric Infrared Sounder (AIRS; Chahine et al., 2006) , the Infrared Atmospheric Sounding Interferometer (IASI; Clerbaux et al., 2009), and the Cross-track Infrared Sounder (CrIS; Han et al., 2013)."

**Comment C1.2 –** L85: methodology used for comparing each instrument IS described

**Reply:** This has been corrected: "…and the methodology used for comparing each instrument is described in Sect. 3."

**Comment C1.3** – L221: add units of CO reference profiles

**Reply:** We have added the units of the CO reference profiles to the text: "First, however, the CO reference profiles (provided in units of mol mol$^{-1}$ with respect to dry air) must be converted to partial columns, and then summed to obtain the total column concentration…"

**Comment C1.4** – L265: null-space error $e_n$ (also known as the smoothing error). Why not prefer the smoothing error? You should add the Rodgers reference. If you want to use the null-space error, you have to explain why. You give references of different studies but we need explanations.

**Reply:** We agree with the reviewer and we have changed all references to "null-space error" in the text to "smoothing error" as this is more common terminology. The notation used to denote this error has also been changed from $e_n$ to $e_{smoothing}$ throughout the text.

**Comment C1.5** – L285: In the following section describes.. Remove "In"

**Reply:** This has been corrected: " The following section describes the methods used to compare the TROPOMI and PEARL-FTS datasets."

**Comment C1.6** – L300: A is the VMR averaging kernel > A is the VMR/VMR averaging kernel (A is unit less). Same thing in L343.

**Reply:** We have implemented this suggestion on Pg. 11, L300: "…where I is the identity matrix, A is the unitless VMR/VMR averaging kernel of the PEARL-FTS…", and on Pg. 12, L343: "ACE-FTS VMR profiles are then smoothed using the VMR/VMR averaging kernel A of the PEARL-FTS using a similar form to Eq. 6…"

**Comment C1.7** – L328-330: What reference should you give when talking about sensitivity ? Rodgers 2000? Rodgers and Connor 2003? As already said, you mention a work published in a paper "following xx et al.". You should be more specific and cite the proper reference for terminology.

**Reply:** We believe Rodgers (2000) is the correct reference for the concept of retrieval "sensitivity", as this concept is introduced on Pg. 46 of Rodgers (2000). We have added a reference to Rodgers (2000) to Pg. 9, L250 where we first define sensitivity: "However, since ACE-FTS performs solar occultation measurements, a sensitivity (i.e., the ratio of information coming from the measurement versus the information from the a priori as defined by Rodgers (2000))".

We have also added a reference to Rodgers (2000) on Pg. 12, L328: "To achieve this, the sensitivity of the retrievals (as defined by Rodgers (2000)) at each level $k$ was first computed by summing the corresponding rows of the averaging kernel matrix, $\sum_i A_{ki}$, following the method of Vigouroux (2008)."

**Comment C1.8 –** L362: SEM. Define (to understand the difference with standard error and standard deviation) and justify your choice for using SEM. Make uniform notation.

**Reply:** We acknowledge that SEM (i.e., standard error of the mean) was not properly defined in the text, and also that using "SEM" in the text might have been confusing and is not a commonly used abbreviation. To make things clearer, SEM was changed to "standard error of the mean" throughout the text to avoid confusion. Pg. 14, L362: "with a small mean bias of $-3.70\times10^{15}\pm1.37\times10^{15}$ molec. cm$^{-2}$ ($-0.68\pm0.25$ %; bias $\pm$  standard error of the mean)…".

Furthermore, to differentiate the standard deviation and the standard errors of the mean, and to explain why both are reported in the comparisons, we have added a sentence to the end of §3.2 briefly describing the differences. Pg. 10, L281: "For each comparison, we provide the standard deviation of the differences $\sigma_{bias}$ as a measure of the spread in the observed differences as well as the standard error of the mean, defined as $\sigma_{bias}/\sqrt{N}$ with $N$ the number of collocations, as a metric for the statistical significance of the reported bias.".

**Comment C1.9** – Caption of Table 3: "standard deviations of the differences" in the first sentence. "standard error of the mean" in the last sentence. It is confusing. The caption should be reformulated.

**Reply:** We have modified the caption of Table 3 to make it less confusing and to differentiate the standard deviations of the differences from the standard errors on the means a bit more clearly: "Summary of the number of collocations, the mean partial column differences, and the standard deviations of the differences between ACE-FTS and TROPOMI globally, and in each latitude region. The relative bias and standard deviation values are computed with respect to ACE-FTS (i.e., 100 x (TROPOMI-ACE-FTS)/ACE-FTS). The uncertainties provided for the absolute and relative biases  are the standard errors on the means."

**Comment C1.10** – L481: coming from the measurement IN this range?

**Reply:** We have implemented this correction on Pg. 22, L481. However, it should be noted that due to a suggestion by reviewer #2, the altitude range of the partial column comparison between ACE-FTS and the PEARL-FTS has now been extended from 10.25-40.17 km to 9.33-66.58 km, which has led to an increase in the partial column DOFS from 0.55 to 0.72. The text now reads as follows (Pg. 22, L480-481): "…however a DOFS of 0.72 implies that there is approximately a third  of the total vertical information coming from the measurement in this range."

**Comment C1.11** – L487: "do not overlap within their bounds of uncertainty on the standard error" Not clear, please reformulate

**Reply:** Similar to our comment above, the altitude range of these partial column comparisons has been extended, and now the results fall within the bounds of standard error on the mean from Griffin et al. (2017), so the original sentence has largely been altered. The new sentence is now (Pg. 23, L486-481): "Although the ACE-FTS and PEARL-FTS retrievals have each been updated in the meantime since this earlier study, the findings presented here are within the range of the standard errors on the mean of those from Griffin et al. (2017) indicating reasonable agreement.".

**References**

Beer, R., Glavich, T. A., and Rider, D. M.: Tropospheric emission spectrometer for the Earth
Observing System's Aura satellite, Appl. Opt., 40, 2356–2367,
https://doi.org/10.1364/AO.40.002356, http://ao.osa.org/abstract.cfm?URI=ao-40-15-2356,
2001.

---

## Author Comment (AC2)

**Response to reviewers**

AMTD Manuscript doi: 10.5194/amt-2021-190 Title: "Inter-comparison of CO measurements from TROPOMI, ACE-FTS, and a high-Arctic ground-based FTS" by Wizenberg et al.

We would like to thank both reviewers for their helpful and thoughtful comments and suggestions. Reviewer comments are in blue, author responses are in black, and any additions to the text are underlined. The line and page numbers provided correspond to the version of the manuscript available on AMTD.

**Response to Reviewer #2**

**Comment C2.1** – Pg. 3, L63: the destriped columns are now available (since processor >02)**

**Reply:** We have modified the text on Pg. 3, L61 to include a mention of this new data product: "Updates to the retrieval spectroscopy and de-striping algorithm methodology proposed in Borsdorff et al. (2019) appear to ameliorate the positive bias observed at high latitude sites-, but this improved CO product is not yet available The public release of this new data product (processor v02.02.00) postdates this analysis, and begins from orbit 19258 on 1 July 2021."

**Comment C2.2 – §2.1: suggest to mention the definition of the TROPOMI grid**

**Reply:** We acknowledge that a description of the TROPOMI vertical grid was missing in §2.1. We have added a sentence to Pg. 4, L106 describing the TROPOMI 50-layer retrieval grid: "The radiative transfer calculations in the retrievals are performed on a 50-layer fixed height vertical grid relative to the topographic surface, typically spanning 0-50km above sea level (Landgraf et al., 2018)."

**Comment C2.3** – Pg 4, L110: "relatively insensitive" ... a bit a confusing phrasing to me, I would say a column avk of 1 means all layers contribute equally?

**Reply:** We agree with the reviewer, and acknowledge that this sentence was confusing. We have modified the text as follows: "In general, for clear-sky retrievals over land, the averaging kernel of the SICOR retrieval is near unity for the entire vertical extent of the profile, meaning that-it is relatively insensitive to the vertical distribution of CO all altitudes contribute equally to the final retrieved column value."

**Comment C2.4** – Pg. 4, L120: as mentioned in the product readme file (PRF) §2 the processors before 010202 contained flaws (e.g., wrongly flagged sunglint pixels). Actually from https://s5pexp.copernicus.eu by combining RPRO and OFFL you can skip any processor below 010202 (see latest report on https://mpc-vdaf.tropomi.eu/ or the PRF on https://sentinel.esa.int/web/sentinel/technical-guides/sentinel-5p/products-algorithms)

**Reply:** We have checked the data used in the comparisons, and the only period in which the processors <01.02.02 were used was the period of 10 November 2017 to 27 November 2017. We have truncated the data at this point to eliminate the bad processor versions from the comparisons. This change only affects the ACE-FTS vs. TROPOMI comparison, and not the PEARL-FTS vs. TROPOMI comparison. We have updated all relevant plots, text and tables to reflect the new time range for the ACE-FTS vs. TROPOMI comparison. We have modified the text on Pg. 4, L118 to remove the mention of the earlier processor versions, and to clarify why these are not being used in the analysis: "In this work, we analyze TROPOMI CO measurements for the period from <del>10</del>-28 November 2017 to 31 May 2020. We use either the reprocessed (RPRO) or offline (OFFL) data files from the most recent processor versions (<del>010001, 010002,</del> 010202, 010300, 010301, and 010302) depending on availability for a given day of observations. Processor versions earlier than 010202 were not used due to wrongly flagged sunglint pixels (Landgraf et al., 2020)."

**Comment C2.5** – §2.2, 2.3: suggest to add uncertainty estimates (similar to §2.1)

**Reply:** For ACE-FTS, a general retrieval uncertainty estimate of 5% is provided in Bernath et al. (2020) (https://doi.org/10.1016/j.jqsrt.2020.107178). We have added a reference to this in the text on Pg. 5, L141: "The VMR profiles are retrieved from the measured infrared spectra using a global-fit algorithm which employs a Levenburg-Marquardt non-linear least-squares fitting method as described in (Boone et al., 2005). For the version 4 ACE-FTS dataset, a general retrieval uncertainty estimate of 5% is provided by Bernath et al. (2020)."

For the PEARL-FTS CO data, the mean retrieval uncertainty (for the full 2006-2020 period) is 2.75%, which we have calculated from the data by adding the systematic and random uncertainty components in quadrature. We have added two sentences to Pg. 6, L180 explaining this: "Additionally, spectroscopic parameters used in the retrieval process for CO are from ATM16 (Toon, 2015), while all other species are from HITRAN 2008 (Rothman et al., 2009). The PEARL-FTS CO retrievals have mean degrees of freedom for signal (DOFS) of 2.2, and a mean retrieval uncertainty of 2.75% over the full 2006 to 2020 time series. This retrieval uncertainty estimate was calculated by adding the systematic and random uncertainty components in quadrature, and it consists of the measurement error (determined from the SNR of the observed spectra), the smoothing error, the spectroscopic line width and line intensity uncertainties from HITRAN, and temperature and solar zenith angle (SZA) uncertainties."

**Comment C2.6** – Pg. 6, L164-165: suggest to use www.ndacc.org and add cams27.aeronomie.be to specify what is meant with the CAMS rapid delivery service

**Reply:** We have added the suggested URLs to the text (Pg. 6, L164-165): "The instrument is part of NDACC (http://www.ndaccdemo.org/about http://www.ndacc.org; De Mazière et al., 2018), and measurements of CO, CH4, and O3 are regularly provided to the Copernicus Atmospheric Monitoring Service (CAMS; http://cams27.aeronomie.be) rapid delivery initiative."

**Comment C2.7** –  $\S2$ : General comment: add more information on measurement characteristics: plots of a typical avk, typical uncertainties, typical dofs, ... this is important to understand some of the arguments further down

**Reply:** We are a little hesitant to add more plots since we feel that there are quite a lot in the manuscript already. However, we have added information on the retrieval uncertainties for the ACE-FTS and PEARL-FTS in §2.2 and §2.3 respectively. We have also added information about the mean DOFS for the PEARL-FTS to the end of §2.3.

**Comment C2.8** – p7, 1205: can you give some information on the typical number of tropomi pixels in a co-location (to know how the threshold of 50 relates to the typical number)?

**Reply:** We have added two sentences to §3.1 to describe the mean number of pixels in the collocations, as well as the number of collocations removed due to this pixel filtering criterion for both the ACE-FTS vs. TROPOMI comparisons and the PEARL-FTS vs. TROPOMI comparisons (Pg. 7, L206): "In the comparisons of ACE-FTS to TROPOMI, the mean number of pixels included in the averages was 11 452, and a total of 1190 collocations were removed due to this pixel filtering criterion. In the comparisons of PEARL-FTS to TROPOMI, the mean number of pixels included in the collocations was 124 858 and only 8 collocations were removed."

**Comment C2.9** – Pg. 8, L215: can you explain what you mean with pixel-to-pixel biases (vs pixel-to-pixel variability)?

**Reply:** We recognize that "pixel-to-pixel biases" was an incorrect choice of words and might imply some type instrument or spectrometer issue. We have removed the mention of pixel-to-pixel biases from the text here (Pg. 8, L215): "For the collocated observations, the mean number of TROPOMI pixels included in the averages was 11 452, indicating that the computed TROPOMI averages are statistically robust, and that pixel-to-pixel variability or biases should be negligible."

**Comment C2.10** – Pg. 8, Eq 1: depending on the VMR profile x being with respect to dry or wet air, this equation will alter. Can you add this specification for x and add a reference to the definition of the column averaged g? The equation is derived from the hydrostatic balance equation and should be (for x being a vmr profile wrt dryair)

 $\sum_{i=1}^{N} \frac{Deltap_i(1-q_i)x_i}{M_{da}g(h_i\lambda)}$

(so no column averaged g but the altitude dependent g and using a specific humidity profile q). Did you approximate  $q \sim 0$ , can you motivate this? Why using the column averaged g instead of g from WGS84 eg?

**Reply:** We have implemented the height-dependent g from the WGS84 reference ellipsoid instead of the column-averaged gravity, which was being used previously. The updated results are now in the paper, and any relevant plots, tables and text have been updated. This change had a relatively small (+0.4%) influence on the bias in the TROPOMI vs. PEARL-FTS comparisons, and a similar influence on the TROPOMI vs. ACE-FTS comparisons.

With regards to the approximation made in Eq. 1, it is identical to that made in appendix A3 of Sha et al. (2021), and had a very small influence in those comparisons (0.2% at the tropical site of Paramaribo). We have expanded upon Eq. 1 to make the approximation that we have made (i.e., q = 0) clearer to the reader:

$$c = \sum_{i=1}^{N} \rho_{da} x = \sum_{i=1}^{N} \frac{(1-q_i) \Delta p_i x_i}{M_{da} g(h_i)} \approx \sum_{i=1}^{N} \frac{\Delta p_i x_i}{M_{da} g(h_i)}$$

We have also added the following sentence to Pg. 8, L226 explaining and justifying this approximation: "In the above equation, due to the lack of H2O profile information in the TM5 priors, we have made the approximation that q = 0 and thus  $\rho_{da} \approx \frac{\Delta p_i}{M_{da}g(h_i)}$ . An identical approximation was made by Sha et al. (2021) who found that this resulted in only a small difference of 0.2% in the bias in comparisons of TROPOMI CO against a ground-based FTS at the tropical site of Paramaribo."

**Comment C2.11** – Pg. 9, Eq 2: same question: x being with respect to dryair will alter the equation and an approximation is used?

**Reply:** We acknowledge that it may be unclear why different equations are used for TROPOMI and ACE-FTS, however this is because ACE-FTS VMR profiles are reported with respect to wet air unlike the TROPOMI a priori's which are with respect to dry-air. We have added a mention

of this on Pg. 9, L235 to clarify the differences: "since the ACE-FTS profiles are reported in VMR units (with respect to wet air), these must converted to partial columns as well."

**Comment C2.12** – Pg. 9, L237: can you explain why the reference profile is used and not the "retrieved profile" (=scaled reference profile)?

**Reply:** We have tested using both the reference profile and the "retrieved" profile to fill the missing altitudes of the ACE-FTS profiles. The difference is almost negligible with a difference of -0.04% in the mean bias globally. We believe it is fine to continue to use the reference profiles instead of the "retrieved" profiles to fill the missing values for now.

**Comment C2.13** – Pg. 10, L269: I assume the true profile in eq 4 is the (unknown) true profile, while in eq 3 it is the ACE-FTS profile? If so, I would suggest to use another label for ACE-FTS in eq 3 (and where applicable).

**Reply:** We are in agreement with reviewer #2 that the notation in Eqs. 3 and 4 was too similar and could cause confusion. We have altered Eq. 3 to make it clearer:

**Old:**  $\rho^{\text{smooth}} = A_{\text{col}}\rho^{\text{true}}$ **New:**  $\rho^{\text{ace}}_{\text{smooth}} = A_{\text{col}}\rho^{\text{ace}}$

Additionally, we have modified the notation in Eq. 2 and any references in the text to match Eq. 3 (i.e.,  $\rho$  was changed to  $\rho^{ace}$ ). This new notation is more consistent with that which is used later on in Eqs. 5 and 6 as well.

Comment C2.14 – p10, l285: ". In the following section describes..." -> "The following..."

**Reply:** We have corrected this in the text: "In-The following section describes the methods used to compare the TROPOMI and PEARL-FTS datasets."

**Comment C2.15** – Pg. 12, L328: sensitivity depends on the AVK units: did you use the AVK acting on vmr profiles (as reported in the GEOMS NDACC files) or the converted AVK acting

on vmr profiles relative to the prior. The latter should be used for the sensitivity (SFIT4 uses relative units).

**Reply:** We have re-created Fig. 1 using the SFIT4 averaging kernels instead of those contained within the GEOMS files. Based on the new Fig. 1, we have decided to extend the altitude range from used in the partial column comparisons between ACE-FTS and the PEARL-FTS from 10.25-40.17 km to 9.33-66.58 km. We have updated the values throughout the text, as well as Fig, 10 to reflect the new altitude range. This change increases the mean DOFS in the altitude range from 0.55 to 0.72. Extending the range any lower towards the surface than this greatly increases the influence of a priori information in the comparisons since the ACE-FTS profiles do not typically extend much lower than 9km on average, and leads to a degraded result in the comparisons.

**Comment C2.16** – Pg. 14, Fig 2: the figure labels (a,b...) doe not correspond to the labels in the legend. The figure (f) has a different scale on the x-axis (1e17) compared to the y-axis (1e18).

**Reply:** The scale of the x-axis in subfigure (f) has been made the same as the y-axis. In the figure caption, panel (f) was incorrectly labeled as (d). We have corrected the figure caption as well: "Correlation plots of collocated ACE-FTS and TROPOMI partial columns in the following latitude bands: (a) Global (90° S to 90° N), (b) N Polar (60° N to 90° N), (c) northern Midlatitudes (20° N to 60° N), (d) Equatorial (20° S to 20° N), (e) southern Midlatitudes (20° S to 90° S), and (d)(f) S Polar (60° S to 90° S). ..."

**Comment C2.17** – Pg. 18, L423: the sign of the null-space also depends on the sign of (I-A)? I would need a bit more information to understand the statement..

**Reply:** We are in agreement with reviewer #2 and believe that they have raised a valid point. The relationship between the sign of the smoothing error and the implications this has for the over/underestimation of the profiles is a bit too convoluted due to the highly variable nature of the TROPOMI column averaging kernels. We have removed the following sentence from Pg. 18, L423: "The negative nature of the null-space errors suggests that the TM5 references profiles are generally underestimating CO concentrations in the vertical extent with respect to the retrieved

ACE-FTS columns.". We have also removed another sentence from Pg. 19, L429 which made a similar claim.

**Comment C2.18** – Pg. 21, Fig 8: suggestion to use the same scales on the x axis for both subfigures: this will allow to see the effect of the smoothing more clearly

**Reply:** We agree with reviewer #2 that the x-axis scales should be the same to better highlight the effect of the smoothing operation. The x-axis scales have been made the same for both subfigures as suggested.

Comment C2.19 – Pg. 22, L479: "mean total column DOFS" -> "mean total DOFS"

**Reply:** We have implemented this correction: "The collocated PEARL-FTS retrievals have a mean total <del>column</del>-DOFS of 2.2 +- 0.37, while in the range of 10.25 to 40.17 km the mean DOFS is 0.55 + 0.24."

**Comment C2.20** – Pg. 24, L511: 'latitudinal dependence ... present in both the unsmoothed and smoothed cases': can you relate this latitudinal dependence to the reported uncertainties? Is it significant?

**Reply:** We have expanded upon this sentence to relate the observed biases to the ACE-FTS retrieval uncertainties, to make it clearer that they are significant (Pg. 24, L511): "Additionally, the latitudinal dependence of the biases is present in both the unsmoothed and smoothed cases, and the magnitude of the observed biases exceeds the ACE-FTS retrieval uncertainties of 5% in all latitude regions except the northern Mid-latitudes, indicating that the observed differences are significant."

**References**

Bernath, P., Steffen, J., Crouse, J., and Boone, C.: Sixteen-year trends in atmospheric trace gases from orbit, J. Quant. Spectrosc. Ra-diat. Transf., 253, 107 178, https://doi.org/https://doi.org/10.1016/j.jqsrt.2020.107178, https://www.sciencedirect.com/science/article/pii/S0022407320302958, 2020.